# ModeRNN: Harnessing Spatiotemporal Mode Collapse in Unsupervised Predictive Learning

## Abstract

Learning predictive models for unlabeled spatiotemporal data is challenging in part because visual dynamics can be highly entangled in real scenes. Due to the interference and competition between the learning of various dynamics modes, we find that most existing approaches often degenerate to learning ambiguous motion patterns and thus producing blurry prediction results. We name this phenomenon *spatiotemporal mode collapse* (STMC) and explore it for the first time in the context of unsupervised predictive learning. The key idea is to provide the model with a strong inductive bias to discover the modular, compositional structures of latent modes. To this end, we propose ModeRNN, which introduces a *decoupling-aggregation* framework to learn structured hidden representations between recurrent states. It first introduces a set of *mode slots* with independent parameters to extract individual components of visual dynamics. Considering that multiple spatiotemporal modes may co-exist in a sequence, we then use learnable importance weights to adaptively aggregate the slot features into a unified hidden representation for recurrent updates. In a series of experiments on large-scale, real-world datasets, ModeRNN is shown to better mitigate the so-called mode collapse and thus further benefit from the learning process on diverse visual dynamics.

## 1 Introduction

Predictive learning is an unsupervised learning paradigm that has shown the ability to discover the *spatiotemporal modes* of visual dynamics (Xu et al., 2019; Goyal et al., 2021). However, for large-scale and real-world datasets (see Figure 1), the modes in visual dynamics can be highly entangled and difficult to learn due to the richness of data environments, the diversity of object interactions, and the complexity of motion patterns. For clarity, in the following discussion, *spatiotemporal modes* are considered to have the following properties:

1. A spatiotemporal mode refers to a representation subspace that corresponds to a family of similar, but not predefined, visual dynamics.

2. Multiple spatiotemporal modes naturally exist in real-world data, even in a single frame.

3. We assume the i.i.d. setup to allow all videos to share the same set of spatiotemporal modes in a dataset. Different data may have different compositional structures over the modes.

Under these assumptions, video prediction models are required to (i) decouple the potentially mixed spatiotemporal modes from raw video frames, (ii) understand the compositional structures on top of the learned modes, and (iii) learn the state transitions based on the compositional structures.

Otherwise, since the learned dynamics with respect to different modes may interfere and compete during training, it remains challenging for the prior art in video prediction to generate less blurry future frames based on an ambiguous understanding of mixed physical processes. We refer to this empirical phenomenon as *spatiotemporal mode collapse* (STMC), which is mainly caused by *the collapse of learned representations into invalid subspaces when compromising to multiple spatiotemporal modes in the training set.* Unlike the widely concerned *mode collapse* problem in generative adversarial networks, STMC has not drawn much attention because predictive learning is supposed to be well constrained by the image reconstruction loss. However, due to the limitation of model size, STMC occurs when the model cannot effectively decouple mixed spatiotemporal modes and infer their

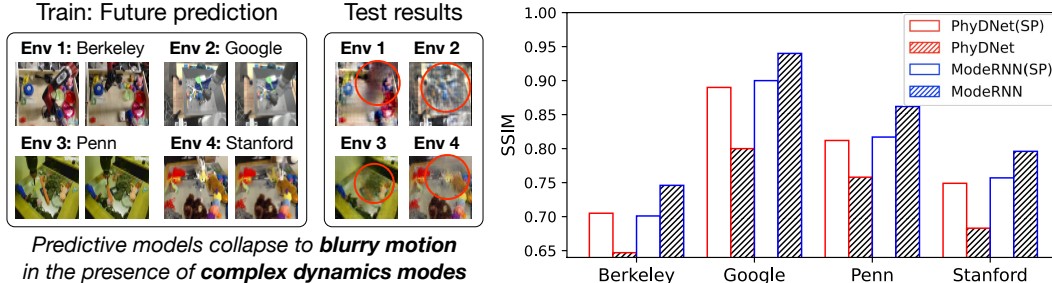

Figure 1: Qualitative and quantitative illustrations of *spatiotemporal mode collapse* on the large-scale, real-world RoboNet dataset. **(Left)** RoboNet contains complicated dynamics modes in videos collected in various environments. The prediction results collapse to blurry motions due to the incompatibility of learning various dynamics modes. **(Right)** Unlike the prior art (Guen & Thome, 2020) that performs better when separately trained in the subset of each environment (denoted as "SP"), the proposed ModeRNN manages to benefit from large-scale learning in all environments.

underlying structures. As a result, its responses to different modes tend to lose diversity and may collapse to a meaningless average of multiple representation subspaces of valid modes.

In Figure 1 (left), we can observe the existence of STMC on a large-scale video dataset named RoboNet (Dasari et al., 2019), in which potential spatiotemporal modes may come from seven different robot platforms (*e.g.*, Baxter and WidowX), four data collection environments (*e.g.*, Berkeley and Stanford), and a variety of unlabeled robot control tasks (*e.g.*, pushing and grasping). An additional outcome of STMC is that we can achieve a performance gain when training individual models in separate subsets with remarkably different visual dynamics, as shown in Figure 1 (right). However, such a dilemma prevents the model from growing into big ones that allow *scalable* training on large-scale, natively multimodal spatiotemporal sequences.

We explore STMC for the first time in *unsupervised* predictive learning. The core idea is to provide a strong inductive bias for the predictive model to discover the compositional structures of latent modes. To this end, we propose ModeRNN, a new modular recurrent architecture that learns structured hidden representations through a set of *mode slots*[1], where each of them responds to the representation subspace of a single spatiotemporal mode. ModeRNN also introduces a *decoupling-aggregation* framework to process the slot features in three stages, which is completely different from existing predictive models with modular architectures (Xu et al., 2019; Goyal et al., 2021).

The first stage is recurrent state interaction and slot binding, in which we use the multi-head attention mechanism (Vaswani et al., 2017) to enable the memory state to interact with the input state and previous hidden state of RNNs. We name the memory state "*slot bus*", because for each sequence, it is initialized from a multi-variate Gaussian distribution with learnable parameters, and thereafter refined using the slot features at each time step. By using the slot bus as the queries, multi-head attention can naturally decouple modular components from hidden representations and bind them to particular mode slots. Features in each slot are then independently modeled using per-slot convolutional parameters. The second stage in each ModeRNN unit is slot fusion, motivated by the assumption that, there can be multiple spatiotemporal modes in a single video and similar videos can be represented by similar compositional structures over the mode slots. Therefore, we assign slot features with learnable importance weights and aggregate them into a unified hidden representation, which is then used in the third stage to update the slot bus and generate the output state of the ModeRNN unit.

We empirically show the existence of STMC on five datasets, and include the results on three real-world datasets in the manuscript, including the large-scale RoboNet dataset that has various data collection environments and multiple robot control tasks, the KTH dataset with six types of human actions that has been widely used by previous literature, and the radar echo dataset for precipitation forecasting that contains time-varying modes of seasonal climates. In addition, we include results on a Mixed Moving MNIST dataset and the Human3.6M dataset in the appendix. In a series of quantitative and visualization results, we demonstrate the effectiveness of ModeRNN in mitigating STMC and learning from highly entangled visual dynamics.

---

[1]The concept of "*slot*" was initially introduced by Locatello et al. (2020) to denote the object-centric features in static scene understanding. We borrow this term here for the subspaces of spatiotemporal representations.

## 2    RELATED WORK

**RNN-based predictive models.**    Many deep learning models based on RNNs have been proposed for spatiotemporal prediction (Ranzato et al., 2014; Srivastava et al., 2015; Shi et al., 2015; Oh et al., 2015; De Brabandere et al., 2016; Villegas et al., 2018). Shi et al. (2015) integrated 2D convolutions into the recurrent state transitions of standard LSTM and proposed the convolutional LSTM (ConvLSTM) network, which can model the spatial correlations and temporal dynamics in a unified recurrent unit. More recent approaches have extended the prediction ability of ConvLSTM in different aspects (Wang et al., 2017; Oliu et al., 2018; Wang et al., 2019b;a; Yao et al., 2020; Guen & Thome, 2020; Yu et al., 2019; Su et al., 2020; Lin et al., 2020; Lee et al., 2021). For example, as an important compared model of our approach, SA-ConvLSTM (Lin et al., 2020) incorporates self-attention in the recurrent state transitions in ConvLSTM to obtain more global context information across time. However, unlike our approach, it does not learn decoupled representations to understand individual components in complex visual dynamics. Besides deterministic models, probabilistic models were proposed to explicitly consider the uncertainty in future prediction (Mathieu et al., 2016; Vondrick et al., 2016; Tulyakov et al., 2018; Xu et al., 2018; Wang et al., 2018; Denton & Fergus, 2018; Castrejon et al., 2019; Kwon & Park, 2019; Bhagat et al., 2020). We use a typical stochastic video generation approach (Denton & Fergus, 2018) based on conditional VAE as a compared model.

**Unsupervised predictive learning for spatiotemporal disentanglement.**    Previous work has focused on learning to disentangle the spatial and temporal features from visual dynamics (Denton et al., 2017; Guen & Thome, 2020; Hsieh et al., 2018; Wu et al., 2021). These methods factorize spatiotemporal data into feature subspaces with strong priors, *e.g.*, assuming that the spatial information is temporally invariant. Another line of work is to learn predictive models for unsupervised scene decomposition such as (Xu et al., 2019; Hsieh et al., 2018). Unlike the above models, our approach uses a set of modular architectures in the recurrent unit to represent the mixed spatiotemporal dynamics. The most relevant work to our method is the *Recurrent Independent Mechanism* (RIM) (Goyal et al., 2021), which consists of largely independent recurrent modules that are sparsely activated and interact via soft attention. ModeRNN is different from RIM in three aspects. First, it is specifically designed to tackle STMC in real-world environments. Second, it learns modular features by incorporating multi-head attention in the recurrent unit, and performs state transitions on compositional features with learnable importance weights. Third, the modular structures in ModeRNN are frequently activated responding to the mixed visual dynamics. ModeRNN is compared with the state of the art in Section 4, including SA-ConvLSTM (Lin et al., 2020), PhyDNet (Guen & Thome, 2020), CrevNet (Yu et al., 2019), RIM (Goyal et al., 2021), and LMC (Lee et al., 2021).

## 3    MODERNN

We propose ModeRNN to reduce *spatiotemporal mode collapse* (STMC) in unsupervised predictive learning. The key idea is to build a *decoupling-aggregation* framework to model the recurrent state transitions of mixed spatiotemporal modes. In this section, we first discuss the basic network components in ModeRNN and then describe the details in the decoupling-aggregation recurrent unit.

### 3.1    MODE SLOTS & SLOT BUS

**Mode slots.**    The decoupling-aggregation framework is built upon a set of hidden representations named *mode slots*. The term *slot* is in part borrowed from previous work for unsupervised scene decomposition (Locatello et al., 2020). We use it here to respond to a family of similar visual dynamics, that is, we aim to bind each mode slot to the representation subspace of each spatiotemporal mode one-to-one. Slot features can be viewed as latent factors that can explicitly improve the unsupervised decoupling of mixed dynamics across the dataset.

**Slot bus.**    Assuming that multiple spatiotemporal modes naturally co-exist in real-world videos, all slots dynamically respond with different importance weights to form compositional representations, which are then used to update a long-term memory state, termed *slot bus*. The hierarchical structure of mode slots and the slot bus leads to a better understanding of the complex and highly mixed dynamic patterns *without mode annotations*. From similar data samples, the model is allowed to learn similar compositional structures over the slots. On the contrary, for distinct visual dynamics, it shows significant differences in the learned importance weights to update the slot bus features. Therefore, it

provides a solution to STMC. Specifically, the slot bus is initialized from a learnable, multi-variate Gaussian distribution, whose mean and variance encode the global priors for the entire dataset.

## 3.2 MODECELL

To learn and leverage the mode slots, we introduce a novel recurrent unit named ModeCell, which follows a decoupling-aggregation framework with three modules, *i.e.*, the state interaction and slot binding module, the adaptive slot fusion module, and the slot bus transition module.

### 3.2.1 STATE INTERACTION AND SLOT BINDING

This module decouples the mixed spatiotemporal modes from raw video frames to mode slots. To achieve this, as shown in Figure 2, it first uses multi-head attention to allow the slot bus to interact with the input and hidden states of the unit, and thereby divides them into separate subspaces. It then binds the features to each mode slot using neural networks with per-slot independent parameters.

Multi-head attention (Vaswani et al., 2017) is widely used in neural language and image processing, and in this work, it is incorporated in the state transitions of ModeRNN. This mechanism allows interactions between the previous slot bus $\mathcal{B}_{t-1}$, the current input state $\mathcal{X}_t$, and the previous hidden state $\mathcal{H}_{t-1}$ (see Figure 2). Formally, at each time step, we first apply 2D convolution projections to $\mathcal{B}_{t-1}$. We then flatten the result to 1D and split it into $N$ mode slots along the channel dimension, such that $\{\text{slot}_{t-1}^1, \ldots, \text{slot}_{t-1}^N\} = \text{Split}(\text{Reshape}(W_{\mathcal{Q}} * \mathcal{B}_{t-1}))$. Note that $\mathcal{B}_{t-1} \in \mathbb{R}^{d_h \times d_w \times (d_x + d_s)}$ and $\text{slot}_{t-1}^n \in \mathbb{R}^{d_h d_w (d_x + d_s)/N}$, where $d_x$ is the channel number of input state, $d_s$ is that of hidden state, and $d_h \times d_w$ indicates the spatial resolution of the slot bus tensor. To improve efficiency, we use two $3 \times 3$ depth-wise separable convolutions (Chollet, 2017) for $W_{\mathcal{Q}}$. We use $\{\text{slot}_{t-1}^1, \ldots, \text{slot}_{t-1}^N\}$ as the queries $\{\mathcal{Q}_t^1, \ldots, \mathcal{Q}_t^N\}$ in multi-head attention, and apply similar operations to obtain keys $\{\mathcal{K}_t^1, \ldots, \mathcal{K}_t^N\}$ and values $\{\mathcal{V}_t^1, \ldots, \mathcal{V}_t^N\}$ based on the concatenation of input state and hidden state, $\mathcal{I}_t = [\mathcal{X}_t, \mathcal{H}_{t-1}]$. We then perform multi-head attention and reshape the $N$ output slot features back to 3D tensors:

$$\text{slot}_t^n = \text{Reshape}\left(\text{softmax}\left(\frac{\mathcal{Q}_t^n \mathcal{K}_t^{n\mathsf{T}}}{\sqrt{d_k}}\right)\mathcal{V}_t^n\right), \ n \in \{1, \ldots, N\}, \tag{1}$$

where $d_k$ is the dimensionality of the key vectors used as a scaling factor.

Multi-head attention brings two benefits to the forward modeling of spatiotemporal data. First, since $\mathcal{B}_{t-1}$ can be unrolled along the recurrent state transition path to be represented as the transformation of slot features at the previous time step, using $\mathcal{B}_{t-1}$ as attention queries allows the model to extract features from $\mathcal{X}_t$ and $\mathcal{H}_{t-1}$ by jointly attending to prior information at different slots. Second, the architecture with $N$ attention heads can naturally help factorize the hidden representation into $N$ subspaces, corresponding to $N$ spatiotemporal modes. The output at each attention head is then updated by a per-slot feed-forward network (FFN) with independent parameters:

$$\text{slot}_t^n = \text{FFN}^n(\text{slot}_t^n) = \max(0, W_{\text{FFN}}^n * \text{slot}_t^n), \ n \in \{1, \ldots, N\}, \tag{2}$$

where $*$ denotes the convolution operator and $W_{\text{FFN}}^n$ are $3 \times 3$ convolution kernels. Through random parameter initialization and stochastic gradient descent, the independent networks $\{W_{\text{FFN}}^1, \ldots, W_{\text{FFN}}^N\}$ would most likely be optimized into parameter subspaces far from each other, thus forcing the slots to bind to various modes in mixed visual dynamics.

### 3.2.2 ADAPTIVE SLOT FUSION

We explicitly consider the co-existence of various modes in a video frame, and use the adaptive slot fusion module to aggregate the decoupled slot features through importance weights. The similarity of visual dynamics is reflected in the similar importance weights of mode slots, while different visual dynamics can be distinguished by different significance of each slot feature. This mechanism prevents ModeRNN from making ambiguous predictions in highly non-stationary data environments.

The implementation of this module is largely inspired by the *mixture of experts* (Shazeer et al., 2017), which introduces the gated networks to control the information flow from base models in an ensemble. We improve the gated architecture to dynamically aggregate decoupled slots $\{\text{slot}_t^1, \ldots, \text{slot}_t^N\}$ with

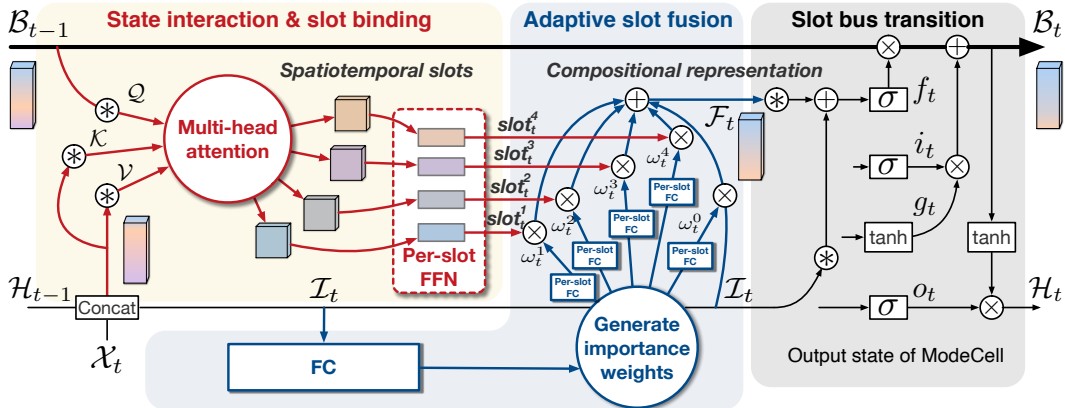

Figure 2: ModeCell tackles spatiotemporal mode collapse via a decoupling-aggregation framework based on mode slots. Multiple ModeCells are stacked to form the complete architecture of ModeRNN.

learnable importance weights $\{\omega_t^1, \ldots, \omega_t^N\}$, and finally have the compositional representation $\mathcal{F}_t$ based on the learned importance weights and corresponding slot features:

$$\omega_t^n = \sigma \circ \text{FC}^n\Big(\text{FC}\big(\frac{1}{d_h \times d_w}\sum_{i=1}^{d_h}\sum_{j=1}^{d_w}\mathcal{I}_t(i,j)\big)\Big), \; n \in \{0, \ldots, N\}, \tag{3}$$

$$\mathcal{F}_t = \omega_t^0 \cdot \mathcal{I}_t + \sum_{n=1}^{N}\omega_t^n \cdot \text{slot}_t^n. \tag{4}$$

In the first line, we use the global average pooling to encode the contextual information, which is the concatenation $\mathcal{I}_t$ of current input state and previous hidden state, into dimensionality $\mathbb{R}^{(d_x + d_s)}$. For memory efficiency, we use a simple fully connected (FC) layer to reduce the dimensionality and get the compact feature in $\mathbb{R}^{(d_x + d_s)/2}$. Then we introduce $(N + 1)$ slot-independent FC layers $\{\text{FC}^0, \ldots, \text{FC}^N\}$ to generate the importance weights for the original input $\mathcal{I}_t$ and each mode $\text{slot}_t^n$, where $\sigma$ denotes the Sigmoid activation function. In the second line, we aggregate all mode slots as well as the input into a compositional representation $\mathcal{F}_t$ based on the learned importance weights.

### 3.2.3 SLOT BUS TRANSITION

The compositional state $\mathcal{F}_t$ builds a hierarchical representation on top of the slot features. We use four sets of $\mathcal{F}_t$ and $\mathcal{I}_t$ to form the input gate $i_t$, forget gate $f_t$, output gate $o_t$, and modulated slot bus input $g_t$. We then update the slot bus state following an LSTM-style recurrent transition mechanism:

$$\begin{pmatrix} g_t \\ i_t \\ f_t \\ o_t \end{pmatrix} = \begin{pmatrix} \tanh \\ \sigma \\ \sigma \\ \sigma \end{pmatrix} \circ \begin{pmatrix} W_g \\ W_i \\ W_f \\ W_o \end{pmatrix} * [\mathcal{F}_t, \mathcal{I}_t], \qquad \mathcal{B}_t = f_t \odot \mathcal{B}_{t-1} + i_t \odot g_t. \tag{5}$$

Finally, we generate the output state of ModeCell as $\mathcal{H}_t = o_t \odot \tanh(\mathcal{B}_t)$. $\mathcal{H}_t$ is taken as inputs by the next ModeCell at the upper level when multiple ModeCells are being stacked in ModeRNN. In other words, ModeCell is to ModeRNN what LSTM is to the stacked LSTM network.

## 4 EXPERIMENTS

We quantitatively and qualitatively evaluate ModeRNN on three real-world datasets in the main paper. We also conduct experiments on the Human3.6M dataset and the Mixed Moving MNIST dataset. Due to page limitation, we include these additional results in Appendix B and Appendix C.

- **RoboNet:** The RoboNet dataset (Dasari et al., 2019) includes more than 15 million video frames collected by 7 different robot arms from 4 environments. It thus contains a large diversity of spatiotemporal modes of rigid motions. We randomly select 4,000 videos for

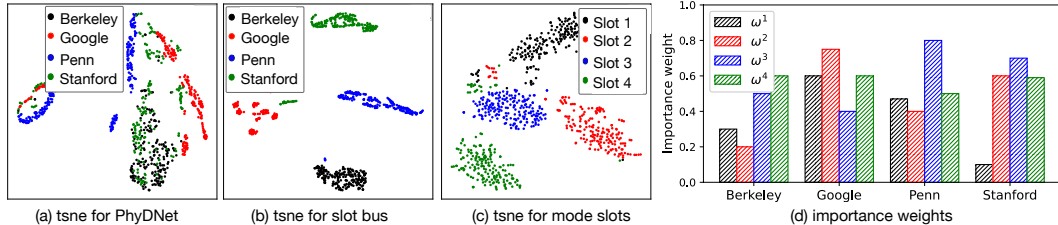

Figure 3: (a) Demonstration of STMC on RoboNet using an existing video prediction model for disentangling visual dynamics (Guen & Thome, 2020). (b) The slot bus in ModeRNN learns distinct representations for samples from different environments. (c) The four slots in ModeRNN learn decoupled features for various spatiotemporal modes. (d) The importance weights of mode slots respond differently to different data environments in RoboNet, *i.e.*, families of similar video sequences.

testing and use the others for training. We follow the action-free and action-conditioned video prediction setups from the work of Babaeizadeh et al. (2018). In the action-free setup, models are trained to predict the next 10 frames from the previous 5 observations. In the action-conditioned setup, models are trained to predict 10 frames based on 2 observations along with the robot action vectors at all time steps. All images are resized to $64 \times 64$.

- **KTH:** The KTH dataset (Schuldt et al., 2004) contains 6 action categories and involves 25 subjects in 4 different scenarios. It thus naturally contains various modes responding to similar motion dynamics. We use person 1-16 for training and 17-25 for testing, resize each frame to the resolution of $128 \times 128$, and predict 20 frames from 10 observations.

- **Radar Echo:** This dataset contains 30,000 sequences of radar echo maps for training, and 3,769 for testing. It naturally contains various spatiotemporal modes of fluid dynamics due to seasonal climate. Models are trained to predict the next 10 radar echoes based on the previous 10 observations. All frames are resized to the resolution of $384 \times 384$.

We train the models with the $L_2$ reconstruction loss and use the ADAM optimizer (Kingma & Ba, 2015) with a starting learning rate of $0.0003$. The batch size is set to $8$, and the training process is stopped after $80,000$ iterations. All experiments are implemented in PyTorch (Paszke et al., 2019) and conducted on NVIDIA TITAN-RTX GPUs. We run all experiments three times and use the average results for quantitative evaluation. Typically, we use $4 \times 64$-channel stacked recurrent units in most RNN-based compared models, including ModeRNN, ConvLSTM (Shi et al., 2015), PredRNN (Wang et al., 2017), SA-ConvLSTM (Lin et al., 2020), and PhyDNet (Guen & Thome, 2020). ModeRNN is also compared with state-of-the-art methods, including SVG (Denton & Fergus, 2018), SAVP (Lee et al., 2018), CrevNet (Yu et al., 2019), RIM (Goyal et al., 2021), and LMC (Lee et al., 2021).

## 4.1 Demonstration of Spatiotemporal Mode Collapse

**STMC occurs on large-scale, real-world datasets.** RoboNet naturally has the label of the data collection environments, including *Berkeley*, *Google*, *Penn* and *Stanford*. To demonstrate that STMC does exist in real-world datasets, and our approach can overcome STMC, we assume that different environments correspond to different combinations of the spatiotemporal modes. As we have seen in Figure 1, training the existing models using data samples from all environments leads to ambiguous predictions of object's future movements; While training separate models on the subset of each environment leads to better overall performance. From these results, we may conclude that previous methods degenerate drastically when using all training samples with mixed visual dynamics. These results perfectly match the t-SNE (Van der Maaten & Hinton, 2008) visualization in Figure 3(a), where the cell output states of PhyDNet (Guen & Thome, 2020) are severely entangled and collapse to less discriminative subspaces. In contrast, from Figure 3(b), the compositional slot bus features in ModeRNN show 4 clusters with clear boundaries, corresponding to four robot environments.

**Does STMC still exist in supervised predictive learning?** One may concern that why not use the environment labels as input to help learning the environment-specific representations. There are two reasons. First, in reality, most spatiotemporal modes are implicit and cannot be pre-defined or annotated, even in RoboNet. Therefore, simply using the sparse labels for the environments or

Table 1: Results on the RoboNet dataset in the action-free setup.

| MODEL | SSIM ($\uparrow$) | MSE ($\downarrow$) | PARAM (MB) | MEM (GB) |
|---|---|---|---|---|
| CONVLSTM (SHI ET AL., 2015) | 0.725 | 133.4 | 8.2 | 2.6 |
| PREDRNN (WANG ET AL., 2017) | 0.787 | 110.9 | 11.8 | 3.5 |
| SVG (DENTON & FERGUS, 2018) | 0.792 | 108.2 | 15.2 | 6.5 |
| SA-CONVLSTM (LIN ET AL., 2020) | 0.753 | 116.5 | 10.5 | 3.4 |
| PHYDNET (GUEN & THOME, 2020) | 0.742 | 122.5 | 14.4 | 4.5 |
| PHYDNET W/ ENVIRONMENT LABEL | 0.750 | 116.9 | 14.4 | 4.5 |
| LMC (LEE ET AL., 2021) | 0.783 | 113.4 | 12.4 | 5.9 |
| CREVNET (YU ET AL., 2019) W/ ST-LSTM | 0.794 | 109.4 | 7.0 | 3.3 |
| **MODERNN** | **0.831** | **91.9** | **6.4** | 3.2 |

Table 2: Results on the RoboNet dataset for action-conditioned video prediction.

| MODEL | SSIM ($\uparrow$) | MSE ($\uparrow$) | TRAINING TIME (H) |
|---|---|---|---|
| PHYDNET (GUEN & THOME, 2020) | 0.813 | 106.2 | 20 |
| SVG (DENG ET AL., 2016) | 0.835 | 99.1 | 23 |
| SAVP (LEE ET AL., 2018) | 0.842 | 96.5 | 25 |
| **MODERNN** | **0.874** | **83.5** | 16 |

robot types cannot completely address the STMC problem. Second, as shown in the 6-th line in Table 1, using the environment labels does not empirically improve the prediction results by a large margin. We here follow the well-established practice in Conditional-GAN (Mirza & Osindero, 2014) to encode a one-hot environment label to PhyDNet (Guen & Thome, 2020).

## 4.2 VISUALIZATION OF REPRESENTATIONS LEARNED BY MODERNN

Besides the visualization of slot bus in Figure 3(b), we testify the mode decoupling ability of ModeRNN by visualizing the slot features in Figure 3(c). We can see that features of the 4 mode slots are clustered into 4 groups, indicating the ability to disentangle various spatiotemporal modes. In Figure 3(d), we use the averaged importance weights $\{\omega_t^n\}_{n=1}^4$ on each slot to analyze how the adaptive slot fusion module works. We can see that different robot environments lead to different dependence distribution over the mode slots.

## 4.3 RESULTS ON THE ROBONET DATASET

**Action-free video prediction.** In Table 1, we show the per-frame quantitative results and computational efficiency for action-free video prediction. As we can see, ModeRNN achieves **state-of-the-art** overall performance with **fewer parameters** compared with existing approaches. It can consistently benefit from training with complex visual dynamics in the entire dataset (see the bar chart in Figure 1). Furthermore, as shown in the first example in Figure 4, ModeRNN is the only method that captures the exact movement of the robot arm, while other models make blurry predictions in the motion area.

**Action-conditioned video prediction.** We also conduct experiments under the action-conditioned setup, encoding the inputs of robot action signals using the action fusion module from PredRNN-V2 (Wang et al., 2021). We mainly compare the performance of ModeRNN with that of SVG (Denton & Fergus, 2018) and SV2P (Lee et al., 2018), which are strong baselines as their effectiveness on RoboNet has been well validated in the prior literature. For these models, we draw 100 prediction samples from the prior distribution given a testing sequence and report the results with the best SSIM scores. From Table 2, ModeRNN achieves the best performance in **the shortest training time**. We show the qualitative results in the second case in Figure 4. With the help of the action inputs, ModeRNN has more accurate predictions about the moving trajectories of the robot arm and the object, while the compared models still suffer from the blur effect.

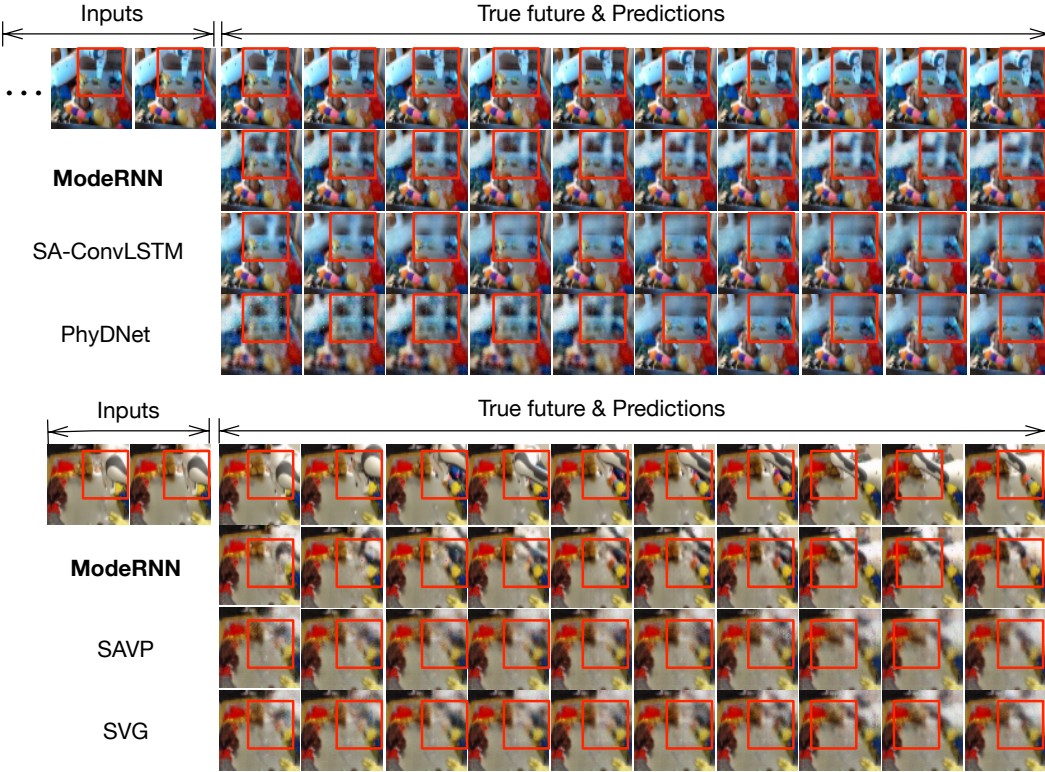

Figure 4: Showcases of future prediction on RoboNet in (**Top**) action-free and (**Bottom**) action-conditioned setups. These examples are randomly sampled from the Stanford environment. We provide examples for other data collection environments in Appendix A.

Table 3: Results on the KTH dataset and the radar echo dataset. For SVG, we report the best results from 100 output samples per input sequence. * indicates the result directly copied from the reference.

| MODEL | KTH | | RADAR | |
|---|---|---|---|---|
| | PSNR (↑) | LPIPS (↓) | CSI30 (↑) | MSE (↓) |
| CONVLSTM (SHI ET AL., 2015) | 24.12 | 0.231 | 0.354 | 97.6 |
| TRAJGRU (SHI ET AL., 2017) | *26.97 | *0.219 | 0.357 | 89.2 |
| PREDRNN (WANG ET AL., 2017) | *27.47 | 0.212 | 0.359 | 84.2 |
| SVG (DENTON & FERGUS, 2018) | 27.73 | 0.196 | - | - |
| CONV-TT-LSTM (SU ET AL., 2020) | 27.59 | 0.198 | 0.363 | 87.6 |
| SA-CONVLSTM (LIN ET AL., 2020) | *29.33 | 0.193 | 0.362 | 86.1 |
| PHYDNET (GUEN & THOME, 2020) | 28.69 | 0.188 | 0.358 | 92.1 |
| CREVNET (YU ET AL., 2019) | 28.82 | 0.183 | 0.381 | 81.5 |
| LMC (LEE ET AL., 2021) | *28.61 | 0.195 | 0.361 | 93.5 |
| **MODERNN** (KTH: $N = 6$; RADAR: $N = 4$) | **29.45** | **0.173** | **0.428** | **65.1** |

## 4.4 RESULTS ON THE KTH DATASET

On this dataset, we use the frame-wise peak signal-to-noise ratio (PSNR) and learned perceptual image patch similarity (LPIPS) (Zhang et al., 2018) as evaluation metrics. We use 6 mode slots in each ModeCell. In the left column in Table 3, we show the quantitative results and find that ModeRNN performs best among all compared methods, including the state of the art proposed in recent two years (Lin et al., 2020; Guen & Thome, 2020; Yu et al., 2019; Lee et al., 2021). We provide the qualitative comparisons in Appendix D, where we observe that ModeRNN can predict the precise position of the moving person. Notably, we also evaluate ModeRNN on a larger human action dataset, Human3.6M, and provide the results in Appendix B.

Table 4: Ablation study of (**Left**) the effectiveness of each model component and (**Right**) the number of mode slots in each ModeCell. Experiments are conducted on the action-free RoboNet dataset.

| MODEL | MSE | MODEL (# MODE SLOTS) | MSE |
|---|---|---|---|
| **MODERNN** | **91.9** | MODERNN ($N = 1$) | 118.1 |
| MODERNN W/O SLOT BINDING | 132.5 | MODERNN ($N = 2$) | 103.4 |
| MODERNN W/O PER-SLOT FFN | 110.7 | MODERNN ($N = 3$) | 94.5 |
| MODERNN W/O ADAPTIVE SLOT FUSION | 121.2 | **MODERNN** ($N = 4$) | **91.9** |
| MODERNN W/O GATED SHORTCUT ($\omega_t^0 \cdot \mathcal{I}_t$) | 128.3 | MODERNN ($N = 5$) | 93.3 |

## 4.5 RESULTS ON THE RADAR ECHO DATASET

Besides the frame-wise MSE, we use the Critical Success Index (CSI) metric, which is defined as $\text{CSI} = \frac{\text{Hits}}{\text{Hits+Misses+FalseAlarms}}$, where hits correspond to the true positive, misses correspond to the false positive, and false alarms correspond to the false negative. A higher CSI indicates better forecasting performance, and it is particularly sensitive to high-intensity echoes. We set the alarm threshold to 30 dBZ for this radar benchmark. As shown in the right column in Table 3, ModeRNN achieves the state-of-the-art overall performance and significantly outperforms the prior art for precipitation forecasting, *i.e.*, TrajGRU (Shi et al., 2017), with a CSI result of 0.428 vs. 0.357 and an MSE result of 65.1 vs. 89.2. We include prediction showcases of the compared models in Figure 14 in Appendix E. Both the quantitative and qualitative results show that ModeRNN effectively learns the non-stationary dynamics of rich spatiotemporal modes from complicated meteorological systems in the real world.

## 4.6 ABLATION STUDY

**Effectiveness of each model component.** In the left column of Table 4, we analyze the efficacy of each component in ModeRNN on the action-free RoboNet dataset and have the following observations. First, removing the slot binding module increases the prediction error by 44.2%, showing the necessity of learning to decouple the dynamics using separate mode slots based on multi-head attention. Second, removing the adaptive slot fusion module increases the prediction error by 31.9%, which strongly demonstrates that it is crucial to learn the state transitions upon the compositional representations based on the slot features. Third, the per-slot FFN in the slot binding module and the gated shortcut ($\omega_t^0 \cdot \mathcal{I}_t$) in the slot fusion module also show significant impact on the final performance. These results verify that parameter isolation is effective in mode decoupling, and reveal the positive effect of an adaptive fusion of rich appearance information and compact spatiotemporal dynamics. Finally, the entire decoupling-aggregation framework that integrates the above techniques in a unified modular model achieves the best performance.

**Number of mode slots.** In the right column of Table 4, we adjust the number of mode slots on RoboNet. We find that the performance first increases rapidly with the growth of the slot number and achieves the best performance at $N = 4$. Notably, using a single slot in each ModeCell achieves similar performance to SA-ConvLSTM (Lin et al., 2020), which incorporates self-attention in the recurrent state transitions but does not have a mode decoupling framework. We perform a similar grid search on other datasets and finally set $N = 6$ on KTH and $N = 4$ on the radar echo dataset.

## 5 CONCLUSION

In this paper, we demonstrated a new phenomenon of spatiotemporal mode collapse (STMC) when training unsupervised predictive models on real-world datasets with highly mixed visual dynamics. Accordingly, we proposed ModeRNN that effectively learns modular features using a set of mode slots. To discover the compositional structures in spatiotemporal modes, ModeRNN adaptively aggregates the slot features with learnable importance weights. Compared with existing models, ModeRNN was shown to prevent the collapse of future predictions, improving qualitative and quantitative results on five datasets. A potential limitation of this work is that, although ModeRNN can be easily generalized as the world model for model-based robot control and has been evaluated on the RoboNet dataset for action-conditioned video prediction, its effectiveness on downstream tasks has not been explored by performing the entire pipeline of model predictive control. We would like to study it in future work.

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

## A    QUALITATIVE RESULTS ON ROBONET FOR DIFFERENT ENVIRONMENTS

We further provide the prediction samples from other environments of RoboNet under both the action-free and action-conditioned setups, including Google, Penn, and Stanford.

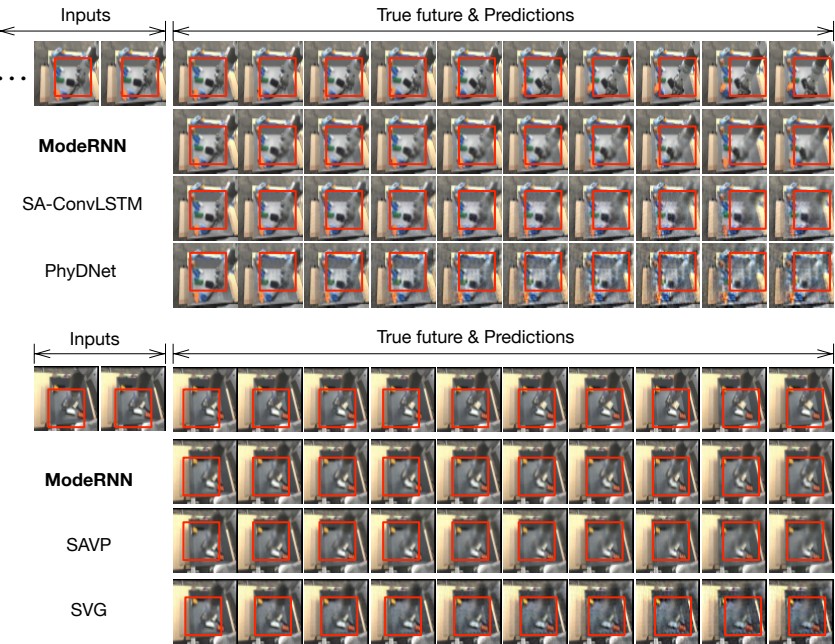

Figure 5: Examples of (Top) action-free and (Bottom) action-conditioned video prediction from the **Google** environment.

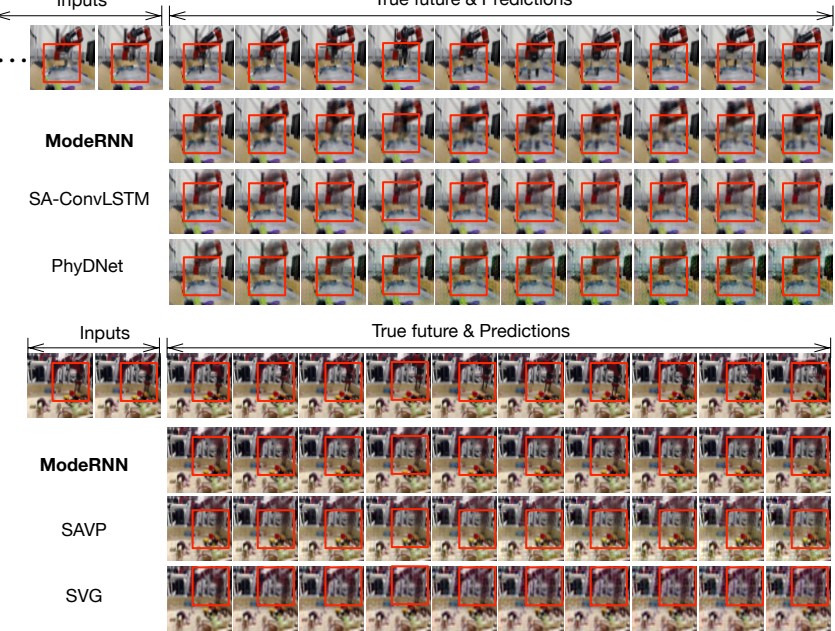

Figure 6: Examples of (Top) action-free and (Bottom) action-conditioned video prediction from the **Berkeley** environment.

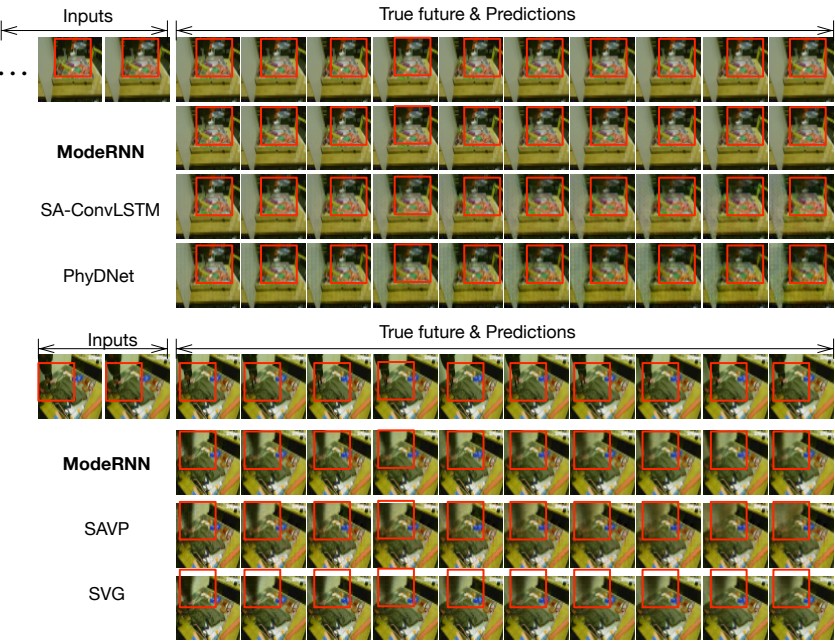

Figure 7: Examples of (Top) action-free and (Bottom) action-conditioned video prediction from the **Penn** environment.

# B   EXPERIMENTS ON THE HUMAN3.6M DATASET

We further conduct experiments on a complex human action dataset, Human3.6M (Ionescu et al., 2013). The Human3.6M dataset contains 2,220 sequences for training, 300 for validation, and 1,056 for testing, involving 17 different scenarios. Compared with KTH, the Human3.6M contains a larger diversity and complexity in spatiotemporal modes, thereby more challenging. We follow the protocol from Wang et al. (2019b) to resize each RGB frame to the resolution of $128 \times 128 \times 3$ and make the model predict 4 future frames based on 4 previous ones. For evaluation metrics, we use the learned perceptual image patch similarity (LPIPS) (Zhang et al., 2018), the frame-wise peak signal-to-noise ratio (PSNR) to evaluate our models. Besides these advanced frame-wise metrics, we also use the Fréchet Video Distance (FVD) (Unterthiner et al., 2018), which is a video-wise metric for qualitative human judgment of generated frame sequences. The FVD can measure both the temporal coherence of the video content and the quality of each frame.

As shown in Table 5, ModeRNN significantly outperforms the previous state-of-the-art method MotionRNN (Wu et al., 2021) (PSNR: 24.2 vs. 22.1, FVD: 16.4 vs. 18.3, LPSIS: 0.123 vs. 0.136). Note that our approach can also obtain great promotion on the FVD metric, which means the prediction results are better in terms of motion consistency. As for the qualitative results, ModeRNN predicts the sharpest sequence compared with other methods and enriches the details for each part of the body, especially for the arms. These results verify the capability of ModeRNN on dealing with diverse and complex spatiotemporal modes in a fully unsupervised way.

Table 5: Quantitative results on the Human3.6M dataset.

| MODEL | PSNR (↑) | FVD (↓) | LPSIS (↓) |
|---|---|---|---|
| SA-CONVLSTM (LIN ET AL., 2020) | 21.3 | 19.2 | 0.153 |
| PHYDNET (GUEN & THOME, 2020) | 22.0 | 18.3 | 0.145 |
| MOTIONRNN (WU ET AL., 2021) BASED ON MIM | 22.1 | 18.3 | 0.136 |
| LMC (LEE ET AL., 2021) | 21.5 | 18.7 | 0.151 |
| **MODERNN** | **24.2** | **16.4** | **0.123** |

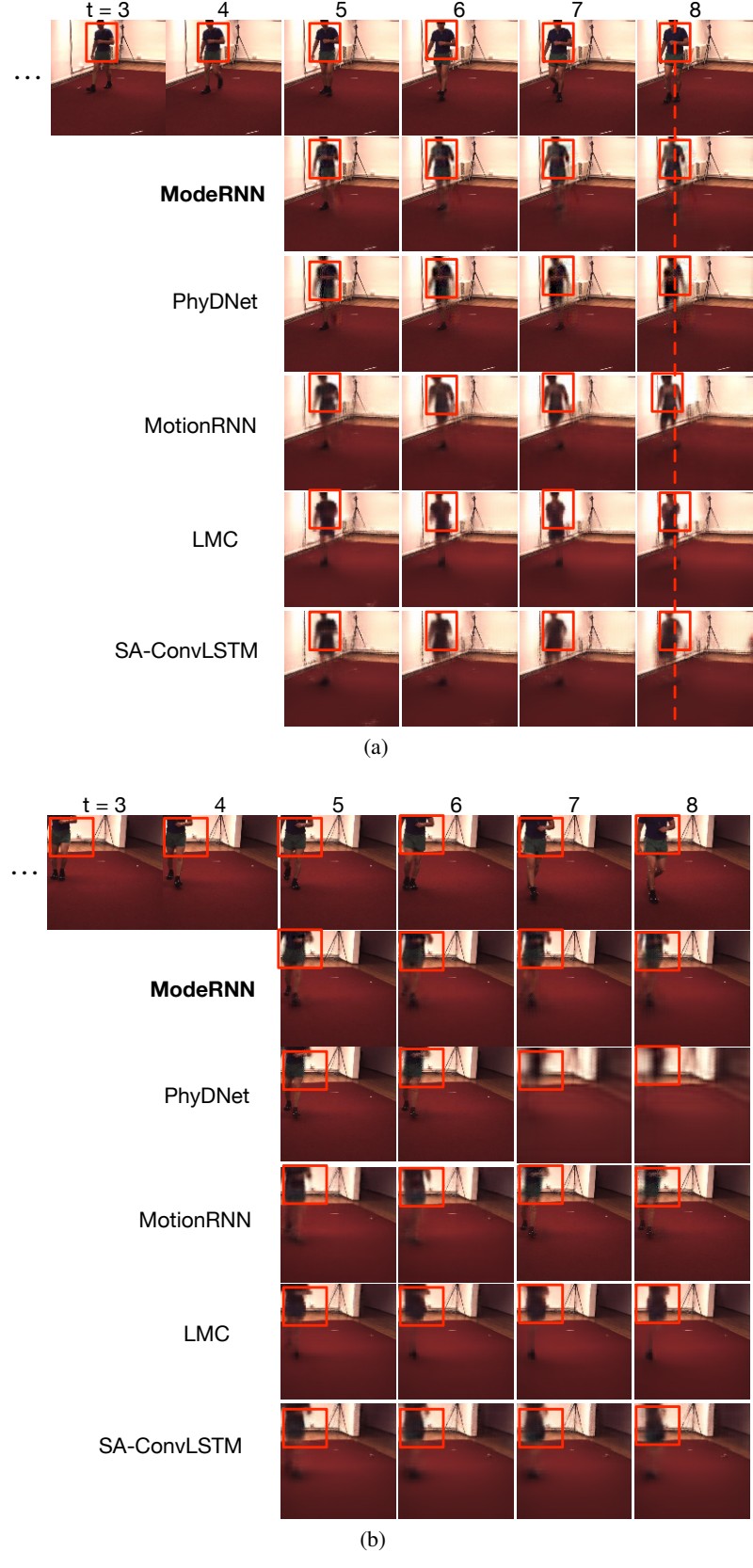

Figure 8: Examples of predicted future frames on the Human3.6M dataset.

## C EXPERIMENTS ON THE MIXED MOVING MNIST DATASET

In the prior works, the number of the flying digits of the Moving MNIST dataset (Shi et al., 2015) is fixed, such as the classic methods ConvLSTM, and recent advanced SA-ConvLSTM, PhyDNet. To approximate the multi-mode phenomenon of the real world, for each sequence, we randomly set the number of the flying digits in the range of 1 to 3. This setup is more challenging than previous convention, which requires the model to handle various spatiotemporal dynamics due to different frequencies of occlusions among sequences. We name this new dataset as the *Mixed Moving MNIST*, which contains 30,000 training sequences, 6,000 validation sequences, and 9,000 testing sequences. Each sequence consists of 20 consecutive frames. The first 10 frames are for the input, and the next 10 frames are for prediction. All the frames are in the resolution of $64 \times 64$.

Table 6: Quantitative results on the Mixed Moving MNIST dataset.

| MODEL | SSIM (↑) | MSE (↓) |
|---|---|---|
| CONVLSTM (SHI ET AL., 2015) | 0.836 | 78.7 |
| PREDRNN (WANG ET AL., 2017) | 0.851 | 67.3 |
| MIM (WANG ET AL., 2019B) | 0.851 | 64.4 |
| RIM (GOYAL ET AL., 2021) | 0.874 | 57.5 |
| SA-CONVLSTM (LIN ET AL., 2020) | 0.854 | 70.3 |
| PHYDNET (GUEN & THOME, 2020) | 0.871 | 60.6 |
| LMC (LEE ET AL., 2021) | 0.856 | 72.5 |
| CREVNET (YU ET AL., 2019) | 0.862 | 58.9 |
| **MODERNN** | **0.898** | **44.7** |

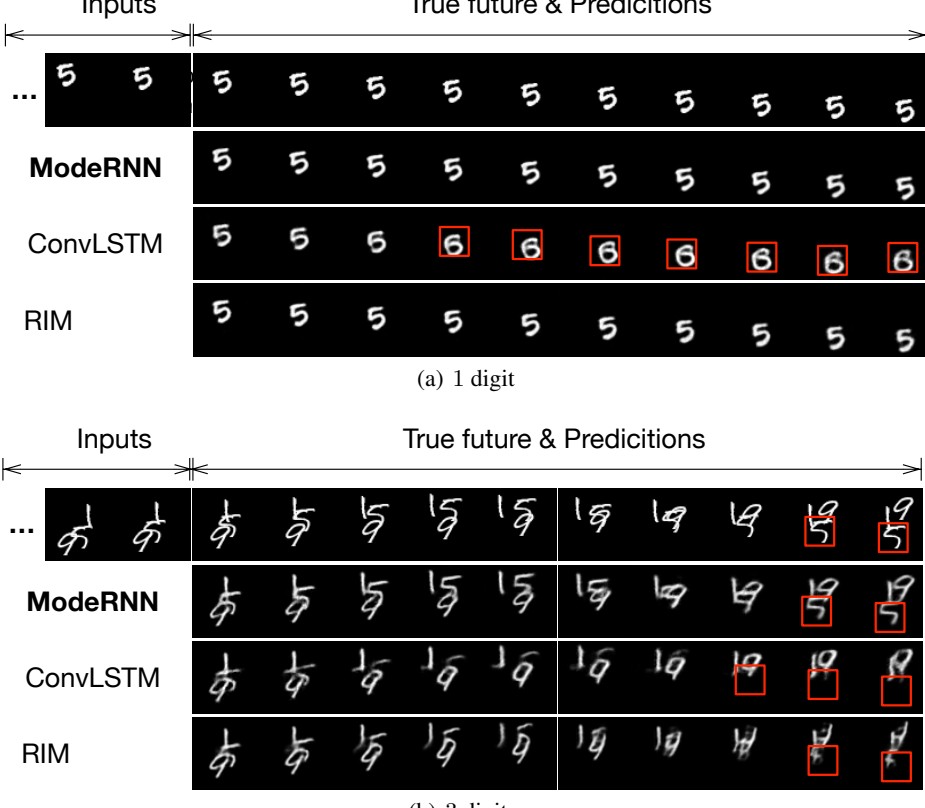

(a) 1 digit

(b) 3 digits

Figure 9: Examples of predicted future frames on the Mixed Moving MNIST dataset.

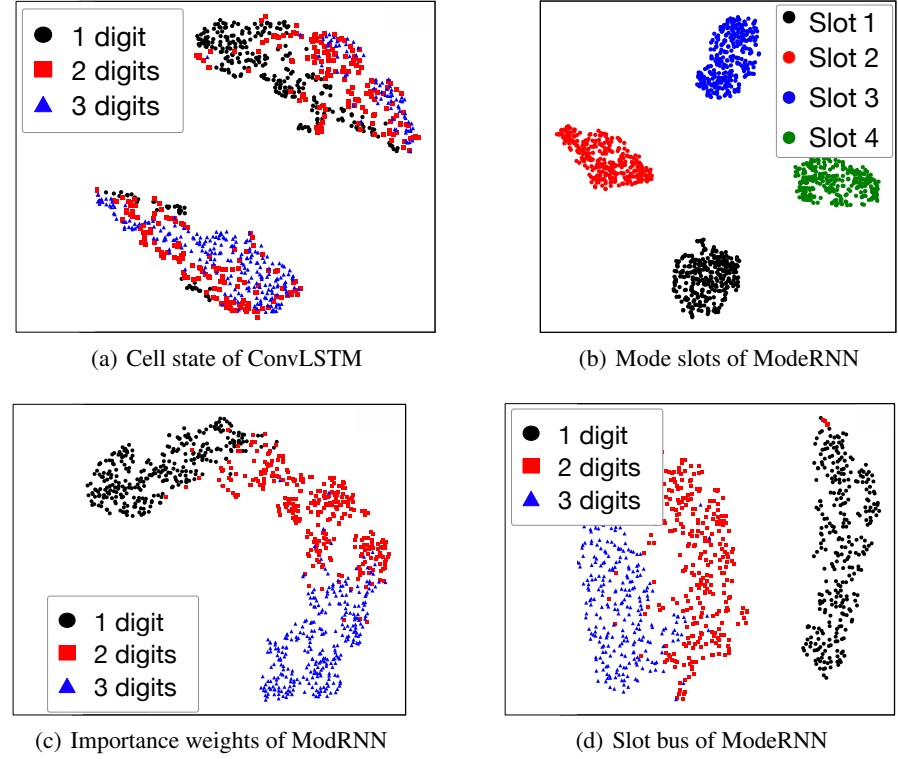

(a) Cell state of ConvLSTM

(b) Mode slots of ModeRNN

(c) Importance weights of ModRNN

(d) Slot bus of ModeRNN

Figure 10: t-SNE visualization on the Mixed Moving MNIST dataset. (a) Illustration of STMC on existing approach. (b) Different mode slots learn different components of visual dynamics. (c-d) The importance weights and slot bus show discriminative representations, which implies the ability to learn less blurry motions and thus alleviates STMC.

In Table 6, we show the overall quantitative results as well as computational efficiency of the compared models on the Mixed Moving MNIST dataset. As we can see, ModeRNN achieves **state-of-the-art** overall performance (SSIM: **0.897**, MSE: **44.7**) compared with existing approaches, including the state-of-the-art approaches proposed in recent two years. Furthermore, as shown in Figure 9, ModeRNN is the only method that can capture the exact movement of each digit, while other models predict the blurry results and the digit "5" is even vanished. All in all, ModeRNN effectively overcomes STMC. It achieves the best performance on the synthetic data which is more difficult than the original Moving MNIST dataset due to a larger variety of visual dynamics.

In Figure 10(a), we visualize the memory state $\mathcal{C}_t$ of ConvLSTM using t-SNE (Van der Maaten & Hinton, 2008)., and find that they are entangled under different digit modes in the Mixed Moving MNIST dataset. It provides evidence that this widely used predictive model cannot learn mode structures effectively. Training the model on a dataset with mixed dynamics leads to severe mode collapse in feature learning, resulting in the entanglement of hidden representations.

## D ADDITIONAL RESULTS ON THE KTH DATASET

We show examples of predicted future frames on the KTH action dataset in Figure 11.

**A-distance.** A-distance (Ben-David et al., 2010) is defined as $d_A = 2(1 - 2\epsilon)$ where $\epsilon$ is the error rate of a domain classifier trained to discriminate two visual domains. In Figure 12(a), we use the A-distance to quantify the STMC in the real-world KTH action dataset. In this experiment, we divide the KTH dataset into two groups according to the visual similarities of human actions. According to the scale of the actions, we can simply group the existing six categories in the KTH dataset into two typical groups:

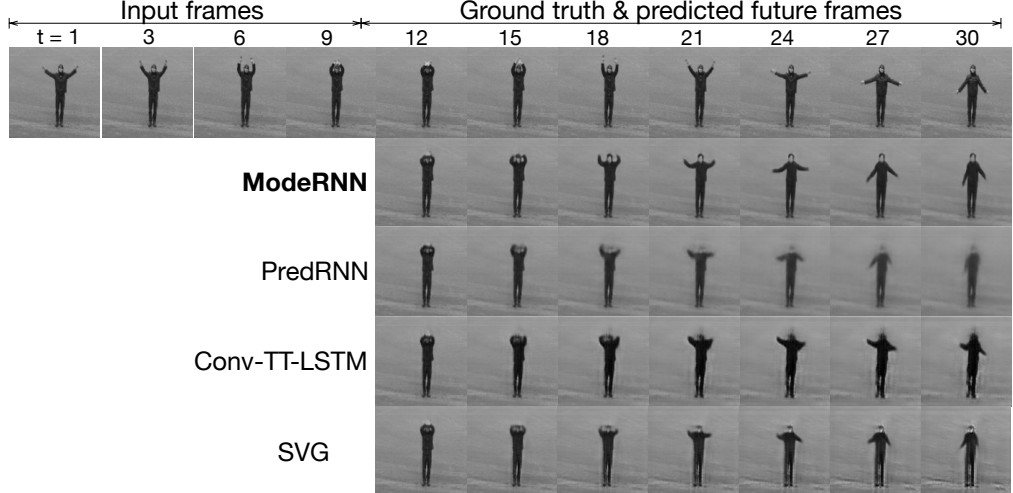

Figure 11: Examples of predicted future frames on the KTH action dataset.

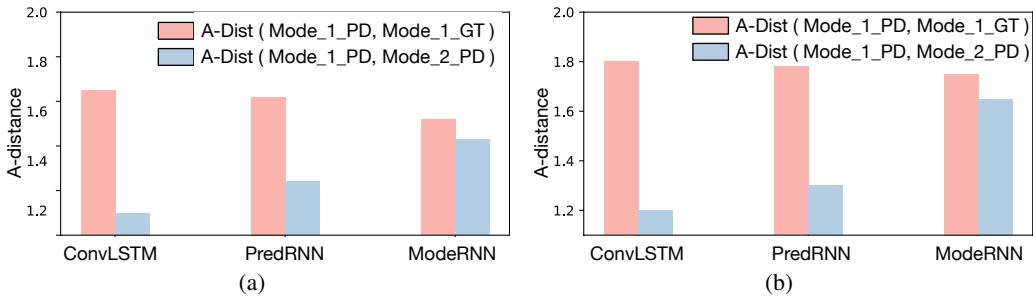

Figure 12: Demonstration of spatiotemporal mode collapse on the KTH dataset using A-distance. (**a**) A-distance based on the memory states, *i.e.*, $\mathcal{B}_t$ for ModeRNN. (**b**) A-distance based on the output states, *i.e.*, $\mathcal{H}_t$ for ModeRNN).

- The first group corresponds to the global movement of the torso, including the categories of running, walking, and jogging.
- The second group corresponds to the local movement of hands, including the categories of hand-clapping, hand waving, and boxing.

We here use the memory state $\mathcal{C}_t$ in ConvLSTM and PredRNN, and the slot bus $\mathcal{B}_t$ in ModeRNN to calculate A-distance. As shown by the blue bars (higher is better), the lower A-distance between the two groups indicates that the learned representations from the two groups are highly entangled. The red bars (lower is better) show the domain distance between features taking as inputs the ground truth frames $\mathcal{X}_t$ and those taking the predictions $\widehat{\mathcal{X}}_t$. STMC happens when the A-distance between predictions of different groups (in blue) becomes much smaller than that between predictions and ground truth (in red).

**t-SNE.** As shown in Figure 13(a), we visualize the memory state of ConvLSTM using t-SNE (Van der Maaten & Hinton, 2008). It is observed that the learned cell states by ConvLSTM are entangled among different action groups. The t-SNE visualization result matches the PhyDNet visualization on the RoboNet dataset shown in Figure 3(a). Thus, these results verify that the STMC also exists under the real-world human motion dataset. While in ModeRNN, we further visualize the learned features of the slot bus in Figure 13(b), which shows 2 clusters with clear boundaries, corresponding to two action groups in the KTH dataset. According to these t-SNE results, we can find that directly training the previous methods on the mixed dynamics will lead to severe STMC in representation learning, shown as the entanglement of hidden representations. These entangled

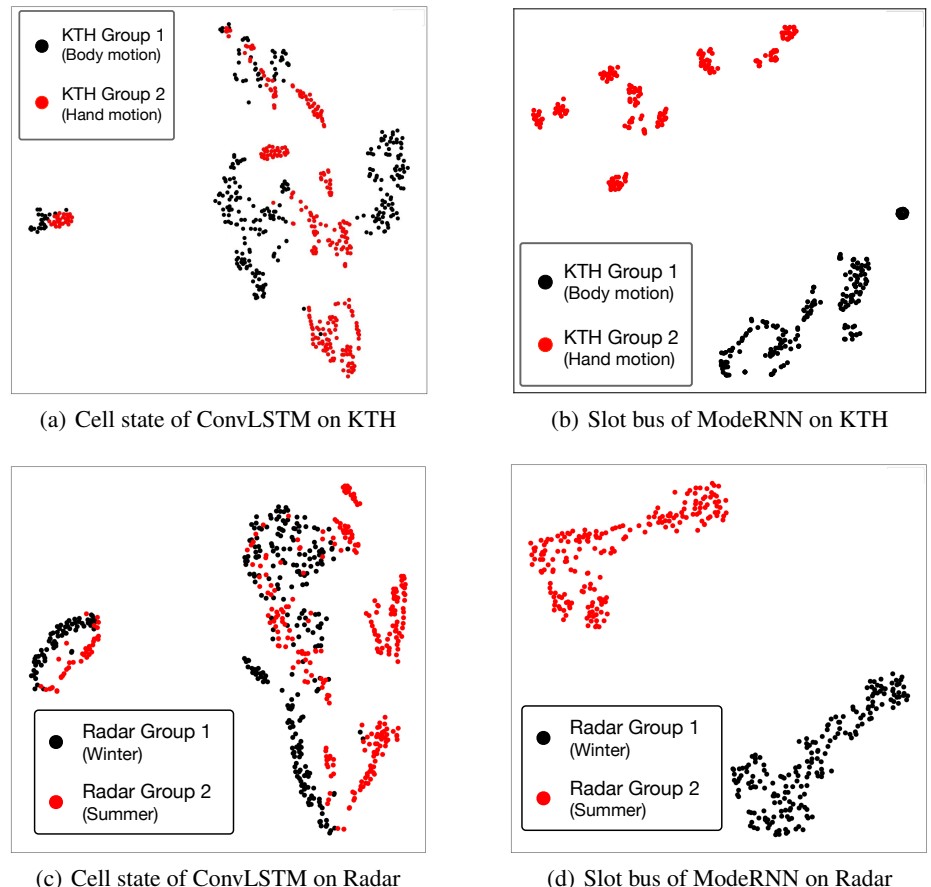

(a) Cell state of ConvLSTM on KTH

(b) Slot bus of ModeRNN on KTH

(c) Cell state of ConvLSTM on Radar

(d) Slot bus of ModeRNN on Radar

Figure 13: (a, c) Illustration of STMC on the existing ConvLSTM model on KTH and radar echo dataset of Guangzhou (GZ). (b, d) The slot bus of ModeRNN shows discriminative representations on different groups of video dynamics. The two groups in KTH respectively correspond to subtle hand motion (*e.g.*, hand-waving, hand-clapping, and boxing) and more global body motion (*e.g.*, running, walking, and jogging). The two groups in Radar are divided by different seasons.

representations will make the model provide a poor ambiguous prediction. In contrast, ModeRNN can effectively overcome the STMC by learning an accurate decoupling for mixed dynamics.

# E  QUALITATIVE RESULTS ON THE RADAR ECHO DATASET

We show examples of predicted future frames on the radar echo dataset in Figure 14.

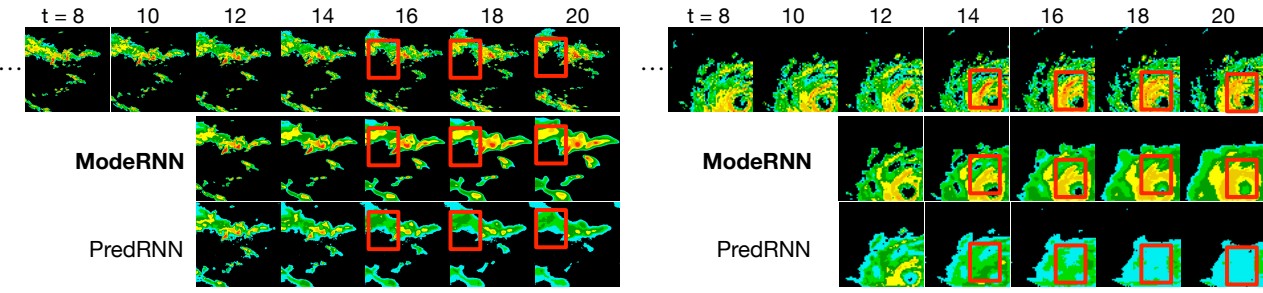

Figure 14: Examples of predicted future frames on the radar echo dataset.

Considering the climate change among different seasons in Guangzhou, we can roughly consider the radar echo dataset into two typical meteorology groups:

- The first group: It corresponds to the windier part of the year, from March to May, with average wind speeds of more than 7.5 miles per hour. There will be drizzles from time to time in these months. We use the radar maps from 2016/3 to 2016/5 and 2017/3 to 2017/4 for training, and use those in 2017/5 for testing.

- The second group: It corresponds to the summer in Guangzhou, which experiences heavier cloud cover, with the percentage of time that the sky is overcast or mostly cloudy is around 80%. We use the radar maps from 2016/6 to 2016/8 and 2017/6 to 2017/7 for training, and use those in 2017/8 for testing.

As shown in Figure 13(c), we also visualize the cell state of ConvLSTM using t-SNEand find that the learned cell state are entangled under different climate groups. It shows that the STMC also exists under the real-world precipitation dataset. In Figure 13(d), we further visualize the features in the slot bus, which show 2 clusters with clear boundaries, corresponding to two climate groups.

## F    FURTHER COMPARISON WITH SA-CONVLSTM

To better position our ModeRNN, we provide a further comparison with the competitive baseline SA-ConvLSTM (Lin et al., 2020) as follows, which combines the self-attention and ConvLSTM to capture the global context information. In the motivation aspect, SA-ConvLSTM does not observe the STMC issue in the large-scale complex dataset, which is a key problem that blocks the predictive model capacity. We demonstrate STMC with extensive visualization and further propose the ModeRNN with full insights to tackle this problem. Technically, there are two core differences between ModeRNN and SA-ConvLSTM:

- SA-ConvLSTM uses self-attention only for the representation aggregation, leading to an inherent lack of the ability to decouple the mixed visual dynamics into several modes. On the contrary, ModeRNN separates the learned representations in several subspaces as *mode slots* and adopts the *slot bus* to connect the decoupled slots along the temporal dimension. This design is highly-motivated by the observation of STMC. As shown in Figure 3 in the main text, the learned mode slots from ModeRNN are clustered into 4 groups. It proves that ModeRNN can ravel out various spatiotemporal modes and handle the mixed dynamics.

- SA-ConvLSTM could not dynamically capture the mixed visual dynamics and adjust to different environments effectively. It only uses the self-attention between recurrent states to capture the global context regardless of various spatiotemporal mode information across different environments. In contrast, ModeRNN develops a modular structure, which adaptively produces compositional features via the *mode slots* and *adaptive slot fusion*.

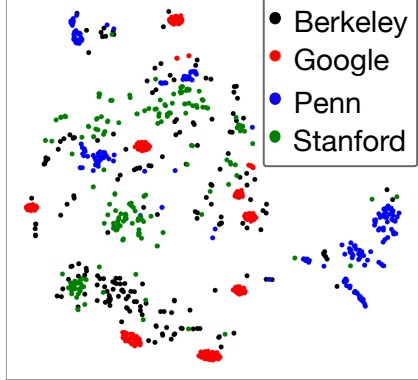

Figure 15: t-SNE visualization of the SA-ConvLSTM hidden states on the RoboNet dataset.

As shown in Figure 15, we further conduct the t-SNE visualization on RoboNet, where the memory states of SA-ConvLSTM are entangled and collapse to the ambiguous representation subspaces, leading to the severe STMC. On the other hand, the quantitative results in RoboNet also show that SA-ConvLSTM does not behave well compared with ModeRNN (SSIM: 0.753 vs. 0.831, MSE: 116.5 vs. 91.9). Thus, SA-ConvLSTM cannot capture mixed visual dynamics to overcome the STMC in a larger complicated real-world dataset.

All in all, our ModeRNN is different from SA-ConvLSTM in both motivation and technical design, which is compared distinctly from the combination of the multi-head attention and ConvLSTM. The carefully designed *mode slots*, *slot bus* and *adaptive slot fusion* form a decoupling-aggregation framework, directly aiming at the STMC, which is the key problem of unsupervised predictive learning. Benefiting from this compact connection between the STMC and model design, ModeRNN achieves the state-of-the-art performance and explainable slot features on extensive datasets with complex spatiotemporal modes.

