# OpenReview forum: "ModeRNN: Harnessing Spatiotemporal Mode Collapse in Unsupervised Predictive Learning"
_ICLR.cc/2022/Conference — ICLR 2022 Submitted_

### Official Review · Reviewer_VMMf · 2021-11-01

**Correctness:** 2
**Technical Novelty And Significance:** 2
**Empirical Novelty And Significance:** 1
**Recommendation:** 5
**Confidence:** 3

**Main Review:**

- The first property mentioned in Introduction section seems a bit simple. Why one would expect each sequence correspond to a single spatio-temporal mode? While this may be true for very simple scenarios, e.g., moving mnist or KTH, for more realistic scenarios, which may not hold.
- Authors made a claim multiple times that they have demonstrated the effectiveness of their approach in dataset with highly mixed visual dynamics. However, I disagree that KTH or Moving MNIST represent such scenarios. Moreover, the training set of such dataset is very small given the complexity of the proposed method.
- The baselines authors compared their approach to don't represent the SOTA or even rather old but strong approaches, such as (Kalchbrenner et al., 2017). Comparing against ConvLSTM or PredRNN is far from enough for ICLR 2022.
- Regarding the method, I'd see it as a good engineering effort of combining existing methods rather than proposing a novel one, with new insights.


**Summary Of The Paper:**

This paper proposes a mechanism to reduce spatiotemporal mode collapse in unsupervised predictive learning. To achieve this goal, the proposed method is built upon the idea that different latent modes in the same data domain should share a set of hidden representation subspaces, which can be represented with various compositional structures based on the features in each subspace. Experimental results show improvement over simpler baselines such as PredRNN or ConvLSTM.

**Summary Of The Review:**

Although this approach seems to be a good engineering effort to come up with an effective model that works reasonably well on very small-scale datasets with relatively low complexity, I believe the novelty of this method is very limited. Moreover, comparison to SOTA and evaluation on larger, complicated datasets are missing.

---

> ### Author Response · Authors · 2021-11-23
> **Our Response to Reviewer VMMf [Part 1]**
>
> Many thanks to reviewer VMMf for the constructive comments.
>
> 1. *Why one would expect each sequence correspond to a single spatiotemporal mode?*
>
> We have carefully rethought the notion of *spatiotemporal mode* and agreed with the reviewer that this statement is not rigorous enough. In the revised draft, we have corrected all relevant statements. In particular with the Introduction section, we rewrite the first paragraph as follows:
>
> The spatiotemporal modes in visual dynamics can be highly entangled and difficult to learn due to the richness of data environments, the diversity of object interactions, and the complexity of motion patterns. Spatiotemporal modes are considered to have the following properties:
> - A spatiotemporal mode refers to a representation subspace that corresponds to a family of similar, but not predefined, visual dynamics.
> - Multiple spatiotemporal modes naturally exist in real-world data, even in a single frame.
> - We assume the i.i.d. setup to allow all videos to share the same set of spatiotemporal modes in a dataset. Different data may have different compositional structures over the modes.
>
> Through the new statements, one would **NOT** expect each sequence to correspond to a single spatiotemporal mode. More precisely, each sequence may be mixed up by a sequence-independent weighted combination from a set of spatiotemporal modes, and each mode may correspond to a compact dynamical component (modeled by the *mode slot*) underlying many sequences. Reasonably, the **decoupling-aggregation** framework proposed with this paper is well fit into the modeling of mixed spatiotemporal dynamics.
>
> 2. *Evaluation on large and complex datasets other than KTH and Moving MNIST.*
>
> We took this concern very seriously and searched actively for more realistic datasets. We came up with two large-scale datasets of complex spatiotemporal dynamics, RoboNet for robot agents and Human3.6M for human beings. Please refer to Sections 4.1-4.3 in the revised draft and Appendix A-B for the detailed results.
>
> We would like to take RoboNet as an example to answer your concern. It includes both action-free and action-condition setups for unsupervised video prediction, while in the latter setup, the video sequences are not only influenced by the environment transitions but also impacted by the robot's actions. As illustrated in the general response, RoboNet contains more than 15 million video frames from 7 types of robotic arms behaved in four visually different environments (Google, Berkeley, Penn, and Stanford). This well reflects the *highly mixed visual dynamics* we emphasize throughout the paper.
>
> Dataset | #Train Seqs | Potential reasons for different spatiotemporal modes
> ---- | --- | ---
> RoboNet | About 160000 | Robots (Baxter, WidowX, ...); Tasks (Pushing, Grasping, ...); Environments (Stanford, Google,...)
>
> We compared ModeRNN with more SOTA methods on RoboNet in both action-free and action-conditioned video prediction setups. We here give a snapshot of the action-free prediction results (full comparisons can be found in Tables 1&2 in the revision). From these results, we can see that ModeRNN remarkably outperforms the SOTA methods.
>
> Method | SSIM (higher is better) | MSE (lower is better)
> ---- | --- | ---
> SA-ConvLSTM (AAAI 2020) | 0.753 | 116.5
> PhyDNet (CVPR 2020) | 0.742 | 122.5
> CrevNet (ICLR 2019)  | 0.794 | 109.4
> LMC (CVPR 2021) | 0.783 | 113.4
> ModeRNN (Ours) | 0.831 | 91.9
>
> Please refer to the "General Response: Large Datasets, Comparison with SOTA, and Novelty" and the revised draft for more details.
>
> 3. *Comparison to SOTA.*
>
> As shown in the general response, we added comparisons with the state-of-the-art video prediction methods SA-ConvLSTM, LMC, CrevNet, and PhyDNet. The full table is included in the revised paper. As shown, ModeRNN significantly outperforms the state-of-the-art methods.
>
> Method | Mixed Moving MNIST ( SSIM/MSE)  |            KTH (PSNR/LPIPS)        |        RoboNet (SSIM/MSE)   |        Radar (CSI30/MSE)
> ---- | --- | --- | --- | ---
> SA-ConvLSTM (AAAI 2020)  |    0.853/70.3     |        29.33/0.196    |       0.753/116.5               | 0.362/86.1
> PhyDNet (CVPR 2020)        |       0.871/60.6      |        28.69/0.188    |     0.742/122.5                  | 0.358/92.1
> CrevNet (ICLR 2019)     |     0.862/58.9        |      28.82/0.183   |      0.794/109.4 |                 0.381/81.5
> LMC (CVPR 2021)        |         0.856/72.5        |      28.61/0.195  |       0.783/113.4|                  0.374/95.8
> ModeRNN (Ours)     |       0.898 /44.7       |      29.45/0.173   |       0.831/91.9   |                0.428/65.1

---

> > ### Author Response · Authors · 2021-11-23
> > **Our Response to Reviewer VMMf [Part 2]**
> >
> > 4. *Novelty.*
> >
> > First up, our paper explores the new spatiotemporal mode collapse (STMC) problem, which has not drawn much attention from the research community of video prediction but is shown practical in large-scale, real-world datasets like RoboNet (Figure 1).
> >
> > Here, we rephrase some important concepts in STMC as follows:
> > - Spatiotemporal mode: It refers to a representation subspace that corresponds to a family of similar, but not predefined, visual dynamics. Nota that: (i) Multiple spatiotemporal modes naturally exist in real-world data. (ii) Different data may have different compositional structures over the spatiotemporal modes.
> > - STMC: The phenomenon that the prediction model degenerates when there are complex spatiotemporal modes in the training set. It is mainly caused by the collapse of learned representations into invalid subspaces when compromising to multiple spatiotemporal modes during training.
> >
> > We summarize the novel contributions in this paper from two aspects:
> > - We found the phenomenon of STMC that most SOTA methods generate blurry prediction results in the presence of complex visual dynamics.
> > - We propose a new **decoupling-aggregation** architecture within the recurrent transitions of off-the-shelf predictive models. The key insight of our approach is to use a modular set of *mode slots* to extract and disentangle the spatiotemporal modes from raw data, and then learn structured representations on top of the mode slots to describe the complex visual dynamics.
> >
> > The connections between our approach and STMC are as follows
> >
> > Data | Model
> > ---- | ---
> > Spatiotemporal mode | A modular mode slot
> > Video sequence with a mixture of modes | The compositional structure over a set of mode slots
> >
> > We agree with the reviewers that ModeRNN borrows existing cutting-edge techniques. However, on one hand, we claim that the use of each model component is fully motivated by our understanding of STMC.
> >
> > On the other hand, we would like to emphasize that these components, including multi-head attention, feed-forward networks, and global average pooling, are originally used for other ML topics. It was still challenging to effectively combine them in a unified video prediction model. In a series of visualizations (Figure 3b-3d) and ablation studies (Table 2), we showed the rationality of each component in the “decoupling-aggregation” framework.

---

### Official Review · Reviewer_Wt6k · 2021-11-03

**Correctness:** 3
**Technical Novelty And Significance:** 2
**Empirical Novelty And Significance:** 1
**Recommendation:** 5
**Confidence:** 4

**Main Review:**

**Strengths:**
* The paper presents a novel temporal model to capture spatiotemporal structures in the data and perform better video prediction. The quantitative experiments were performed on three datasets, mixed moving MNIST, KTH action, and Radar Echo datasets.

**Weaknesses:**
* The cell of the proposed ModeRNN is quite similar to an RNN cell with multi-head attention. One relevant work using a similar idea was not cited. Further, it performs better or on par with the proposed approach in the KTH action dataset.
   Lin et al., "Self-Attention ConvLSTM for Spatiotemporal Prediction," AAAI 2020, https://doi.org/10.1609/aaai.v34i07.6819

* In generative models, mode collapse means the generated samples being identical or very similar to each other. It may be caused by the imbalanced distribution of the training set or dependent structures and bias in the data. The motivation of spatiotemporal mode collapse is valid, however, the examples and used datasets are not enough to represent. For instance, in moving MNIST or other datasets do not contain any imbalanced or entangled factor of variation that will cause mode collapse.

* Even though it was argued on the contrary in the related work, the proposed work is highly related to feature disentanglement. PhyDNet (Guen & Thome 2020) is cited but did not included in the moving MNIST or other dataset results.

 * How were the previous methods in Table 2 and 4 compared? Did you retrain/implement or they were taken from the reference papers? There are several confusing points: for instance, PredRNN's MSE on Radar dataset was reported 44.2 in the reference paper, but it is84.2 in Table 4.

* What is "mixed moving MNIST"? In reference works, mostly moving MNIST (either freshly rendered or the version in Shrivastava et al. 2015) is being used.

* Several typo issues in the entire text (for instance, all titles "Resuls"->"Results").


**Summary Of The Paper:**

This paper defines a phenomenon, spatiotemporal mode collapse in the training of unsupervised predictive models. They propose an RNN-based approach to learning structural hidden representations in temporal data. The proposed idea was experimented with and compared with respect to the convolutional LSTM baseline and several other temporal modeling methods (i.e., RIM or Conv-TT-LSTM).

**Summary Of The Review:**

I think the proposed problem of spatiotemporal mode collapse was not covered and described clearly. This makes it difficult to understand where the contribution of the proposed method comes from and what it improves. Spatiotemporal slots idea reminds the attention modules on the temporal data and leads to a weighted fusion of temporal weights. It is very similar to self-attention ConvLSTMs. Mainly the introduced problem is not clear and also approach is limited in novelty. These are the points affected my decision.

---

> ### Author Response · Authors · 2021-11-23
> **Our Response to Reviewer Wt6k  [Part 1]**
>
> Thank you very much for the constructive comments.
>
> 1. Summary Of The Review: *Mainly the introduced problem is not clear and also approach is limited in novelty.*
>
> We clarify the spatiotemporal mode collapse (STMC) problem as follows:
>
> As shown in Figure 1 in the revision, STMC is a new and practical phenomenon in large-scale, real-world datasets like RoboNet. It describes the learning difficulty in the presence of highly entangled visual dynamics in the training set.
>
> Here, we rephrase some important concepts in STMC as follows:
> - Spatiotemporal mode: It refers to a representation subspace that corresponds to a family of similar, but not predefined, visual dynamics. Nota that: (i) Multiple spatiotemporal modes naturally exist in real-world data. (ii) Different data may have different compositional structures over the spatiotemporal modes.
> - STMC: The phenomenon that the prediction model degenerates when there are complex spatiotemporal modes in the training set. It is mainly caused by the collapse of learned representations into invalid subspaces when compromising to multiple spatiotemporal modes during training.
>
>
> Our paper explores STMC for the first time in the research field of video prediction. We here summarize the novel contributions in this paper from two aspects:
> - We found the phenomenon of STMC that most SOTA methods generate blurry prediction results in the presence of complex visual dynamics.
> - We propose a new **decoupling-aggregation** architecture within the recurrent transitions of off-the-shelf predictive models. The key insight of our approach is to use a modular set of *mode slots* to extract and disentangle the spatiotemporal modes from raw data, and then learn structured representations on top of the mode slots to describe the complex visual dynamics.
>
> The connections between our approach and STMC are as follows
>
> Data | Model
> ---- | ---
> Spatiotemporal mode | A modular mode slot
> Video sequence with a mixture of modes | The compositional structure over a set of mode slots
>
> We agree with the reviewers that ModeRNN borrows existing cutting-edge techniques. However, on one hand, we claim that the use of each model component is fully motivated by our understanding of STMC.
>
> On the other hand, we would like to emphasize that these components, including multi-head attention, feed-forward networks, and global average pooling, are originally used for other ML topics. It was still challenging to effectively combine them in a unified video prediction model. In a series of visualizations (Figure 3b-3d) and ablation studies (Table 2), we showed the rationality of each component in the “decoupling-aggregation” framework.
>
> 2. Comparison with SA-ConvLSTM.
>
> Thanks for bringing this relevant work to us, and we have added this paper to the discussion of related work in the revision.
>
> We argue that our approach is very different from SA-ConvLSTM except for the use of an attention mechanism in an RNN architecture. First, from the aspects of motivation and overall framework, ModeRNN tackles the entanglement of spatiotemporal modes in complex visual dynamics. Therefore, it presents a novel **decoupling-aggregation** framework within the state transition of the recurrent unit. As mentioned above, the key insights of the framework are
> - It uses the modular **mode slots** to learn and disentangle the spatiotemporal modes from raw videos.
> - It learns structured representations based on the mode slots for the forward modeling of the complex visual dynamics.
>
> Second, from the aspect of model details,
> - ModeRNN uses multi-head attention for the reason that it naturally provides a way to separate the entangled spatiotemporal modes, and bind them to individual representation subspaces.
> - Existing video prediction models, including SA-ConvLSTM and E3D-LSTM, use self-attention without multiple heads. Therefore, they are unlikely to decouple the mixed visual dynamics.
>
> Third, from the aspect of empirical results, ModeRNN outperforms SA-ConvLSTM due to the contributions of the slot-based, decoupling-aggregation framework.
>
> - RoboNet
>
> Method  | SSIM (higher is better)   | MSE (lower is better)
> --- | --- | ---
> SA-ConvLSTM | 0.753   | 116.5
> ModeRNN (#Slots=4) | 0.831 | 91.9
>
> - KTH
>
> Method  | PSNR (higher is better)   | LPIPS (lower is better)
> ---- | --- | ---
> SA-ConvLSTM | 29.33  | 0.196
> ModeRNN (#Slots=6) | 29.45 | 0.173
>
> On RoboNet, which is a large-scale dataset with more complex visual dynamics (please see our general response on the new dataset), ModeRNN outperforms SA-ConvLSTM by a large margin. On KTH, ModeRNN outperforms SA-ConvLSTM with a comparable model size (we here use 6 mode slots in each ModeCell). For other datasets, please refer to the revision for the full comparisons of these two models on Human3.6M, Radar Echo, and Mixed Moving MNIST.

---

> > ### Author Response · Authors · 2021-11-23
> > **Our Response to Reviewer Wt6k [Part 2]**
> >
> > 3. *Mixed Moving MNIST.*
> >
> > Typically, the classic / SOTA methods ConvLSTM, SA-ConvLSTM, and PhyDNet use the standard training setup on the Moving MNIST dataset, in which the number of the flying digits is fixed. While in our paper, each video sequence is initialized with a random number of digits, ranging from 1 to 3. No labels are provided and all compared models are trained in a fully unsupervised manner.
> >
> > Obviously, the new dataset is more challenging, because the model needs to handle various visual dynamics due to different frequencies of occlusions. To improve the prediction results, it requires the model to better understand the basic components in different motions.
> >
> > In Figure 9(b) in Appendix C, we can see that the digit "5" is lost in the predicted frames of ConvLSTM and RIM, which is probably caused by STMC that the learning process on the 3-digits cases is compromised by the existence of 2-digits cases in the same training set.
> >
> > 4. *(1) The examples and used datasets are not enough to represent STMC. (2) Results on the imbalanced Moving MNIST dataset.*
> >
> > First, we clarify that STMC is a phenomenon that the learned representations collapse into invalid subspaces when compromising to multiple spatiotemporal modes in the training set. It is shown to be a practical issue for various video prediction models on different datasets:
> > - RoboNet: As shown in our General Response for the RoboNet dataset, we showed that STMC does exist on large-scale, real-world datasets. Existing approaches (even SOTA methods) generate blurry prediction results when trained with a large variety of visual dynamics (Figure 1 in the revision).
> > - Mixed Moving MNIST: As shown in our previous response (Q3), we explained the settings of the Mixed Moving MNIST and discussed how the results on this dataset are influenced by STMC, i.e., *the learning process on the 3-digits cases is compromised by the existence of 2-digits cases in the same training set.*
> > - Furthermore, the t-SNE visualizations in Figure 3 (RoboNet), Figure 10 (Mixed Moving MNIST), and Figure 13 (KTH and Radar) confirm that STMC is a real problem on various datasets.
> >
> > Second, as suggested by the reviewer, we build the Imbalanced  Moving MNIST dataset and compare ModeRNN with SOTA methods.  According to the number of flying digits, this dataset can be divided into 3 subsets with imbalanced data volumes:
> >
> > Dataset      | 1-digit (#TrainSeqs/#TestSeqs)     | 2-digits   |  3-digits
> > ---- | --- | --- | ---
> > Mixed Moving MNIST      | 10000/3000    | 10000/3000   | 10000/3000
> > Imbalanced Moving MNIST   | 10000/3000 | 10000/3000   |  5000/3000
> >
> > The results on the Imbalanced Moving MNIST dataset are shown as follows:
> >
> > Method | SSIM (higher is better) | MSE (lower is better)
> > ---- | --- | ---
> > SA-ConvLSTM (AAAI 2020) | 0.773 | 80.2
> > PhyDNet (CVPR 2020) | 0.789 | 76.9
> > LMC (CVPR 2021) | 0.786 | 79.7
> > CrevNet (ICLR 2019) | 0.792 | 74.5
> > ModeRNN | 0.812 | 57.1
> >
> > From the above results, we can see that ModeRNN significantly outperforms other SOTA methods.
> >
> > 5. *Comparison with PhyDNet.*
> >
> > We here compare the quantitative results of ModeRNN with PhyDNet on all five datasets. Full quantitative and qualitative comparisons have been included in the revision.
> >
> > - RoboNet:
> >
> > Method  | SSIM (higher is better)   | MSE (lower is better)
> > ---- | --- | ---
> > PhyDNet |     0.742 | 122.5
> > ModeRNN |  0.831 | 91.9
> >
> > - KTH:
> >
> > Method  | PSNR (higher is better)   | LPIPS (lower is better)
> > ---- | --- | ---
> > PhyDNet  | 28.69        | 0.188
> > ModeRNN | 29.45 | 0.173
> >
> > - Radar:
> >
> > Method  | CSI30 (higher is better)   | MSE (lower is better)
> > ---- | --- | ---
> > PhyDNet |     0.358 | 92.1
> > ModeRNN |  0.428 | 65.1
> >
> > - Human3.6M:
> >
> > Method  | PSNR (higher is better)   | FVD (lower is better) | LPIPS (lower is better)
> > ---- | --- | ---  | ---
> > PhyDNet  | 22.0  |  18.3 | 0.145
> > ModeRNN | 24.2 | 16.4 | 0.123
> >
> > - Mixed Moving MNIST:
> >
> > Method  | SSIM (higher is better)   | MSE (lower is better)
> > ---- | --- | ---
> > PhyDNet |     0.871 | 60.6
> > ModeRNN |  0.898  | 44.7
> >
> > 6. *(1) Retrained or taken from the reference? (2) The results of PredRNN on the Radar dataset.*
> >
> > First, we retrain all compared models on RoboNet, Radar, Human3,6M, and Mixed Moving MNIST using the provided source code. For KTH, we use $\ast$ to indicate the results directly taken from the reference paper.
> >
> > Second, the Radar dataset in our paper is different from that in the work of PredRNN: (i) They have different training/test splits; (ii) They have different spatial resolutions, i.e. 384*384 (ours) vs 100*100 (PredRNN).
> >
> > 7. *Typo issues.*
> >
> > We have corrected the typos and uploaded the revised paper.
> >
> > __

---

### Official Review · Reviewer_W8mj · 2021-11-03

**Correctness:** 4
**Technical Novelty And Significance:** 4
**Empirical Novelty And Significance:** 4
**Recommendation:** 8
**Confidence:** 3

**Main Review:**

Strong points:
- Overall well motivated novel architecture for addressing the issue of mode collapse in unsupervised predictive training.
- The proposed decoupling-aggregation framework is demonstrated to learn a highly diverse set of mode-factors which are combined to cover a diverse set of output modes with high accuracy.
- The analytical experiments show that the performance of ModeRNN consistently (across datasets) benefits from increased diversity of spatio-temporal modes, whereas the performance of other methods often diminishes when trained on more diverse data.
- The description seems sufficiently detailed for reproducing the experiments.

Comments/Questions:
1. Inconsistencies between text, equations and figure:
- Eq. 3 is missing the shared FFN for dimensionality reduction
- Text should explicitly state that the slot bus input $g_t$ is tanh activated unlike the gates. I’d also name it something like slot bus input instead of input modulation “gate”, since the input gate $i_t$ is usually considered to do the modulation of the cell input.
2. Motivation for implementation details in adaptive slot fusion missing.
3. For clarity: Is $\sigma(I_t) \cdot (W_\text{fuse}^0 * I_t)$ what you call the residual connection? I’m not exactly sure about the terminology, but I believe residual connections are usually linear.
4. I think instead of $slot_{t-1}^*$, you could just write $Q_{t}^*$, since these queries do not correspond to the spatio-temporal slots of the previous time-step. Using the same notation is confusing.
5. Since the figure doesn’t have a descriptive caption, I’d consider placing it on the same page as the description of the architecture.
6. A question about the ModeRNN (i.e. the stacked ModeCell): Do the higher layers receive only the cell output $\mathcal{H}_t$ as input or do they also have access to the slot bus $\mathcal{B}_t$?
7. Typo: Figure 5: Ture should be True
8. Did you also calculate the A-distance using the cell outputs instead of the memory states?
9. The t-SNE visualization of Figures 3 and 5 for KTH and the Radar Echo dataset would be interesting. Are the modes as clearly separated?
10. There’s a broken figure reference in Appendix B.3
11. I might have overlooked it, but I didn’t see a comparison of ModeRNNs with different numbers of ModeCells. I can imagine you played a bit with this hyperparameter, I’d be interested to know your observations.
12. The method for selecting hyperparameters is not described.
13. How do training times compare to ConvLSTM, RIM, and other architectures?
14. No discussion of limitations
15. Do you have initial results on datasets with a larger diversity of spatio-temporal modes (e.g. Human3.6M or KITTI)?

I have updated my score as my concerns are mostly addressed.

**Summary Of The Paper:**

The manuscript proposes a novel architecture with a slot-based decoupling-aggregation framework for unsupervised sequence prediction. The model is motivated by preventing spatio-temporal mode collapse, which affects many existing methods. The experiments clearly show that the model addresses this issue and performance comparisons across three commonly used datasets are presented.

**Summary Of The Review:**

Overall, interesting work. However, for stronger recommendation of acceptance some of the points above should be addressed.

---

> ### Author Response · Authors · 2021-11-23
> **Our Response to Reviewer W8mj**
>
> Thank you very much for the encouraging and constructive comments.
>
> 1. *Missing equation term and other issues.*
>
> As suggested by the reviewer, we have already fixed the missing equation term (Q1), the name of $g_t$ (Q1, changed to *modulated slot bus input*), the notations of queries (Q4), the position of Figure 2 (Q5), the typo (Q7), and the figure reference (Q10). Please see the revision for details.
>
> 2. *Motivation for implementation details in adaptive slot fusion.*
>
> The adaptive slot fusion module is a key component in the **decoupling-aggregation** framework. For the motivation of its implementation details, we have:
> - Global average pooling (Eq. 3): used as the well-established fusion method to encode the contextual information.
> - The slot-shared FC layer (Eq. 3): to reduce the dimensionality and get the compact feature.
> - Slot-independent FC layers (Eq. 3): to generate the importance weights for individual mode slots, as well as for the gated connection from the input to improve the information flow.
> - Aggregation (Eq. 4): to generate structured representations based on the decoupled slot features, which are then used by the slot bus transition module to describe the complex visual dynamics.
> - The gated shortcut connection (Eq. 4): to enable the final output of this module to compromise between the richness of the appearance information in the input tensor and the compact states of spatiotemporal modes.
>
> 3. *Clarity for "residual connection".*
>
> As described above, we use the gated shortcut connection from the input to *enable the final output of this module to compromise between the richness of the appearance information in the input tensor and the compact states of spatiotemporal modes*.
>
> 6. *Do the higher layers receive only the cell output or do they also have access to the slot bus?*
>
> As described in Section 3.2.3 in the revision, *$\mathcal H_t$ is taken as inputs by the next ModeCell at the upper level when multiple ModeCells are being stacked in ModeRNN. In other words, ModeCell is to ModeRNN what LSTM is to the stacked LSTM network*.
>
> 8. *Did you also calculate the A-distance using the cell outputs instead of the memory states?*
>
> As suggested, in Figure 12 in Appendix D, we added the A-distance results based on the cell output $\mathcal H_t$ in the bottom layer in ConvLSTM, PredRNN, and ModeRNN. Similar to the results based on the memory states, they further demonstrate the existence of the spatiotemporal mode collapse (STMC) in real-world data.
>
> 9. *The t-SNE visualization for KTH and Radar Echo dataset.*
>
> We added these t-SNE results in Figure 13 in the Appendix. As shown:
> - STMC still exists for KTH and Radar Echo.
> - The slot bus of ModeRNN learns discriminative representations on different groups of video dynamics (KTH: body motion vs. hand motion; Radar Echo: winter vs. summer).
>
> 11. *Comparison of ModeRNNs with different numbers of ModeCells.*
>
> We use the RoboNet dataset to study the effect of using different numbers of ModeCells.
>
> #ModeCells | MSE (lower is better)
> ---- | ---
> 1 | 117.2
> 2 | 106.7
> 3 | 99.3
> 4 | 91.9
> 5 | 93.2
> 6 | 94.2
>
> As shown in the Table above, the ModeRNN with 4 ModeCells performs best on RoboNet. We used the same setup on other datasets.
>
> 12. *How to select the hyperparameters?*
>
> There are not many hyperparameters to tune in ModeRNN, except for the number of ModeCells and the number of mode slots in each ModeCell. Basically, we set the hyperparameters that can lead to a comparable model size with the SOTA methods. For the number of mode slots, we follow the common practice of previous literature to use the validation set to tune the hyperparameters on RoboNet, Human3.6M, and Mixed Moving MNIST. For KTH, we follow the previous literature of SVG, PredRNN, and SA-ConvLSTM to tune the hyperparameters on the test set.
>
> 13. *Training time.*
>
> We reported the whole training time for each compared method in Table 2 in the revised paper. The entire training procedure of ModeRNN on action-conditioned RoboNet lasts about 16 hours, which is less than that of PhyDNet (20h), SVG (23h), and SAVP (25h). Besides, the number of parameters and memory usage shown in Table 1 also illustrate the efficiency of our approach.
>
> 14. *limitation of ModeRNN.*
>
> Please refer to the conclusion part of the revised paper.
>
> 15. *Results on Human3.6M.*
>
> As suggested, we compared ModeRNN with the following SOTA methods on Human3.6M.
>
> Method  | PSNR (higher is better  | FVD (lower is better) | LPIPS (lower is better)
> ---- | --- | --- | ---
> SA-ConvLSTM [2] | 21.3         | 19.2             |   0.153
> PhyDNet [3] | 22.0    | 18.3     | 0.145
> MotionRNN (MIM)  | 22.1     | 18.3     | 0.136
> LMC [4] |  21.5      | 18.7    | 0.151
> ModeRNN | 24.2   |16.4   | 0.123
>
> We added these results to Appendix B. Also, please check the quantitative comparisons in Figure 8 (Page 15).

---

### Author Response · Authors · 2021-11-23
**General Response: Revision Uploaded**

We thank all reviewers for their constructive comments. Accordingly, we have revised the paper completely and updated the new version (**revision ratio: 50% for experiment and 30% for the rest of the paper**). Please check out the following changes  (highlighted in green color) in the revised paper:
1. We have compared with more SOTA video prediction models, including
- SA-ConvLSTM (AAAI 2020),
- PhyDNet (CVPR 2020),
- LMC (CVPR 2021),
- CrevNet (ICLR 2019),
- MotionRNN (CVPR 2021).
Quantitative results are shown in Tables 1-3 (on RoboNet, KTH, and Radar). Qualitative comparisons are shown in Figure 4. More empirical comparisons are included in the appendix.
2. We added both quantitative and qualitative results on a large-scale, real-world dataset, i.e., RoboNet, which contains complex videos collected from a wide range of robotics environments. Results can be found in Section 4.3 and Appendix A.
3. We added the results on the suggested Human3.6M dataset. Results can be found in Appendix B.
4. In the introduction section, we refreshed Figure 1 with a real case on the large-scale RoboNet dataset. Starting by this new example, on Page 1, we further explained
- what spatiotemporal mode refers to;
- what spatiotemporal mode collapse (STMC) refers to;
- why it is important in practical applications.
5. For clarity, we rename the proposed *spatiotemporal slot* as *mode slot*. Because, as described in Section 3.1 (Mode Slot & Slot Bus), the mode slot is *to respond to a family of similar visual dynamics, that is, we aim to bind each mode slot to the representation subspace of each spatiotemporal mode one-to-one*.
6. In Section 3.2 (ModeCell), we explained the motivation for implementation details of the adaptive slot fusion module. We updated Figure 2 to correspond it more clearly to the main text.
7. In Figure 3(c,d), we included more visualizations to analyze how the mode slots work in the *decoupling-aggregation* framework, that
- Figure 3(c): Different mode slots learn different components in visual dynamics.
- Figure 3(d): The importance weights show discriminative representations for different groups of video sequences.
We added similar t-SNE visualizations on the KTH and Radar datasets in Figure 13 in the appendix.
8. In Table 4, we included more detailed ablation studies of each model component in the *decoupling-aggregation* framework in ModeRNN.
9. In Appendix F, we added detailed comparisons with SA-ConvLSTM (AAAI 2020) in motivations and architecture designs.
10. We moved some contents in the original paper to the appendix in the revision:
- Appendix C: The experiment results on the Mixed Moving MNIST dataset
- Appendix D: the A-distance results and the prediction examples on KTH
- Appendix E: the prediction examples on Radar.

If there are any additional comments on the revision, please do not hesitate to let us know. We are glad to answer any further questions.

---

### Author Response · Authors · 2021-11-23
**General Response: Large Datasets, Comparison with SOTA, and Novelty [Part 1]**

We thank all reviewers for their instructive comments. In addition to the specific response below, here we summarize the response to some common concerns. Corresponding changes have also been included in the revision.

**1. We compared ModeRNN with more advanced video prediction models on larger and more complex datasets.**

We agree with the reviewers that it is important to use a larger dataset to show the existence of spatiotemporal mode collapse (STMC). To this end, we added experiments on the RoboNet dataset [1] in Section 4. Here is a comparison of the three datasets included in the revision:

Dataset | #Train Seqs | Potential reasons for different spatiotemporal modes
---- | --- | ---
RoboNet | About 160000 | Robots (Baxter, WidowX, ...); Tasks (Pushing, Grasping, ...); Environments (Stanford, Google,...)
KTH | About 8500 | Action categories (Running, Boxing, Walking, ...)
Radar |  About 30000 | Seasonal climates (Summer, Winter, ...)

As shown, RoboNet contains a larger diversity of spatiotemporal modes as the videos are collected with different robot platforms and from a wide range of  environments.

We compared ModeRNN with more SOTA methods on RoboNet in both action-free and action-conditioned video prediction setups . We here give a snapshot of the action-free prediction results (full comparisons can be found in Tables 1&2 in the revision). From these results, we can see that ModeRNN remarkably outperforms the SOTA methods.

Method | SSIM (higher is better) | MSE (lower is better)
---- | --- | ---
SA-ConvLSTM [2] | 0.753 | 116.5
PhyDNet [3] | 0.742 | 122.5
LMC [4] | 0.783 | 113.4
CrevNet [5] | 0.794 | 109.4
ModeRNN | 0.831 | 91.9

Furthermore, we presented the following quantitative and qualitative results on RoboNet in the revision:
- Figures 1&3 show the existence of STMC on the SOTA approach, and how the proposed decoupling-aggregation framework in ModeRNN copes with this problem.
- Figure 4 compares the predicted frames of ModeRNN, SA-ConvLSTM, PhyDNet, SAVP, and SVG. It shows that ModeRNN captures the exact movement of the robot arm, while other models make more blurry predictions in the motion area.
- Table 4 investigates the effectiveness of each model component in ModeRNN and the number of mode slots on the RobotNet dataset.

We also included comparisons with SOTA methods on other four datasets, including the suggested Human3.6M dataset (Appendix B).

[1] Sudeep Dasari, et al. "Robonet: Large-scale multi-robot learning." arXiv preprint arXiv:1910.11215 (2019).

[2] Zhihui Lin, et al. "Self-attention convlstm for spatiotemporal prediction." AAAI 2020.

[3] Vincent Le Guen, and Nicolas Thome. "Disentangling physical dynamics from unknown factors for unsupervised video prediction." CVPR 2020.

[4] Sangmin Lee, et al. "Video Prediction Recalling Long-term Motion Context via Memory Alignment Learning." CVPR 2021.

[5] Wei Yu, et al. "Efficient and information-preserving future frame prediction and beyond." ICLR 2020.

---

> ### Author Response · Authors · 2021-11-23
> **General Response: Large Datasets, Comparison with SOTA, and Novelty [Part 2]**
>
> **2. We clarified the novelty of this paper.**
> First up, our paper explores the new spatiotemporal mode collapse (STMC) problem, which has not drawn much attention from the research community of video prediction, but is shown practical in large-scale, real-world datasets like RoboNet (Figure 1).
>
> Here, we rephrase some important concepts in STMC as follows:
> - Spatiotemporal mode: It refers to a representation subspace that corresponds to a family of similar, but not predefined, visual dynamics. Note that: (i) Multiple spatiotemporal modes naturally exist in real-world data. (ii) Different data may have different compositional structures over the spatiotemporal modes.
> - STMC: The phenomenon that the prediction model degenerates when there are complex spatiotemporal modes in the training set. It is mainly caused by the collapse of learned representations into invalid subspaces when compromising to multiple spatiotemporal modes during training.
>
> We summarize the novel contributions in this paper from two aspects:
> - We found the **phenomenon of STMC** that most SOTA methods generate blurry prediction results in the presence of complex visual dynamics.
> - We propose a new **decoupling-aggregation** architecture within the recurrent transitions of off-the-shelf predictive models. The key insight of our approach is to use a modular set of *mode slots* to extract and disentangle the spatiotemporal modes from raw data, and then learn structured representations on top of the mode slots to describe the complex visual dynamics.
>
> The connections between our approach and STMC are as follows
>
> Data | Model
> ---- | ---
> Spatiotemporal mode | A modular mode slot
> Video sequence with a mixture of modes | The compositional structure over a set of mode slots
>
> We agree with the reviewers that ModeRNN borrows existing cutting-edge techniques, which we have dedicated sufficient acknowledgement. However, on one hand, we claim that the use of each model component is fully motivated by our understanding of STMC.
>
> On the other hand, we would like to emphasize that these components, including multi-head attention and feed-forward networks, are originally used for other ML topics. It was still challenging to effectively combine them in a unified video prediction model. In a series of visualizations (Figure 3b-3d) and ablation studies (Table 2), we showed the rationality of each component as well as the “decoupling-aggregation” framework.

---

### Decision · Program_Chairs · 2022-01-20

**Decision:**

Reject

**Comment:**

The paper presents a novel architecture, ModeRNN, for unsupervised video prediction by learning spatiotemporal attention in the latent subspace (slots).  ModeRNN effectively learns modular features using a set of mode slots and adaptively aggregates
the slot features with learnable importance weights. The paper has promising results on several benchmark video prediction datasets.

During the post-rebuttal discussion, the reviewer Wt6k and VMMf responded to the authors' rebuttal, but there was no discussion among them. The consensus is that even though the paper is a very strong engineering effort, it was not clear how the proposed architecture addresses the spatiotemporal mode collapse problem.  T-SNE in Fig. 3/10/13 is insufficient to show disentangled feature space. In fact, PhyDnet was designed to disentangle different factors (physical vs unknown), hence not a good baseline.  [Hsieh et al 2018] is a better fit. In addition, synthetic data examples would be helpful to explain the underlying mechanism of the model and provide more insights for the video prediction community.

Based on this reason, I recommend rejecting this paper as it is now and encourage the authors to revise the draft and submit to future venues.

Hsieh, J. T., Liu, B., Huang, D. A., Li, F. F., & Niebles, J. C. (2018, January). Learning to Decompose and Disentangle Representations for Video Prediction. In NeurIPS.